# Continuous multiplexed population representations of task context in the mouse primary visual cortex

Márton Albert Hajnal [1,7], Duy Tran[2,3,7], Michael Einstein[2], Mauricio Vallejo Martelo [2], Karen Safaryan[2], Pierre-Olivier Polack [4,8], Peyman Golshani [2,5,6,8] ✉ & Gergő Orbán [1,8] ✉

Effective task execution requires the representation of multiple task-related variables that determine how stimuli lead to correct responses. Even the primary visual cortex (V1) represents other task-related variables such as expectations, choice, and context. However, it is unclear how V1 can flexibly accommodate these variables without interfering with visual representations. We trained mice on a context-switching cross-modal decision task, where performance depends on inferring task context. We found that the context signal that emerged in V1 was behaviorally relevant as it strongly covaried with performance, independent from movement. Importantly, this signal was integrated into V1 representation by multiplexing visual and context signals into orthogonal subspaces. In addition, auditory and choice signals were also multiplexed as these signals were orthogonal to the context representation. Thus, multiplexing allows V1 to integrate visual inputs with other sensory modalities and cognitive variables to avoid interference with the visual representation while ensuring the maintenance of task-relevant variables.

To survive and thrive, organisms need to respond in distinct ways to identical stimuli, depending on the wider behavioral context. How the brain represents and maintains these contextual variables during decision making is poorly understood. Task context can be considered as the variable that determines the contingencies between stimulus and reward, but these variables may not be directly observed; rather animals are required to infer their value and store them internally. Several studies have demonstrated contextual modulation of individual neuronal activity patterns in prefrontal[1] and primary sensory cortical areas[1-3]. Yet, it is not known how primary sensory cortical representations are organized such that neurons can encode sensory and cognitive variables simultaneously. Organization of the representation is crucial since the primary sensory area is the gateway to all other downstream areas and therefore compromising sensory information could interfere with processes unrelated to the task. Furthermore, task execution also relies on variables other than task context. Recent studies have demonstrated that non-sensory, task-related variables are represented in the primary visual cortex of rodents[4-7]. Since population responses undergo task-related transformations when performing a visual task[8,9] it is critical to understand how the visual cortical representations contribute to performing multiple tasks and how the V1 representation integrates all task-relevant variables simultaneously.

[1]Department of Computational Sciences, Wigner Research Center for Physics, Budapest 1121, Hungary. [2]Department of Neurology, David Geffen School of Medicine, University of California, Los Angeles, Los Angeles, CA 90095, USA. [3]Albert Einstein College of Medicine, New York, NY 10461, USA. [4]Center for Molecular and Behavioral Neuroscience, Rutgers University, Newark, NJ 07102, USA. [5]Integrative Center for Learning and Memory, Brain Research Institute, University of California, Los Angeles, Los Angeles, CA 90095, USA. [6]West Los Angeles VA Medical Center, CA 90073 Los Angeles, USA. [7]These authors contributed equally: Márton Albert Hajnal, Duy Tran. [10]These authors jointly supervised this work: Pierre-Olivier Polack, Peyman Golshani, Gergő Orbán. ✉e-mail: pgolshani@mednet.ucla.edu; orban.gergo@wigner.mta.hu

Highly structured activity is present in the visual cortex and in particular in V1 even in the absence of visual stimuli[10,11]. Previous studies have been divided with respect to the functional interpretation of this structured activity. On one hand, off-stimulus activity has recently been reported to reflect movement-related variables[12]. This movement-related activity was shown to persist through stimulus and reside in a subspace that is orthogonal to that occupied by visually evoked activity, sharing a single dimension of activity with visual stimulus evoked activity, thus ensuring that processing of visual information remains relatively intact by ongoing movements. On the other hand, statistical structure of neural activity in the absence of stimulus have been related to the subject's prior expectations of future sensory inputs[13] and maintenance of such a prior contributes to inferring environmental features underlying sensory experiences[14,15]. Similar to these environmental features, task context is also inferred from and affects the interpretation of sensory stimuli. While task context has been identified in neural activity both during stimulus presentation and off-stimulus periods[1], it remains an open question, if context is also represented in an independent fashion during stimulus presentation and in intertrial intervals, when stimulus is absent.

To answer these questions, we designed a paradigm where mice switched between two tasks during a single recording session. During both tasks, mice were presented with the same visual and auditory stimuli. The two tasks differed in the stimulus modality (audio or visual) on which the animals needed to base their decisions to obtain rewards. We performed multi-channel electrophysiological recordings in V1 of mice during task performance. Critically, in our paradigm, identical stimuli led to different behavioral outcomes depending on context. Hence, the latent variable related to the context (whether the animal based its choices on visual or auditory input) could be identified in the neuronal population activity. We addressed two questions: First, we investigated the geometry of the representation of the latent variable 'task context' during on-stimulus and off-stimulus activities and the relationship of this cognitive variable to task execution. Second, we investigated how a population of neurons can accommodate multiple task-relevant variables without interference between representations. We found that context can be decoded with high accuracy from V1 population activity both during visual stimulus presentation and inter-trial intervals. This context representation was orthogonal to the representation of visual stimulus identity. Remarkably, relevance of the context representation could be demonstrated by a stronger context-related signal during periods of high behavioral performance. The context representation was persistent across stimulus presentation and off-stimulus periods, yet the representation was not static but displayed distinctive dynamics upon stimulus onset and offset. Other task-relevant variables, identity of auditory stimulus and the choice animals were making, were simultaneously represented in the V1 population in distinct subspaces of the neural activity. Performing two movement related controls, we found that context could be decoded independent of the speed and of posture changes of the animal. In summary, activity patterns in V1 independently represented visual stimuli and cognitive variables relevant to task execution.

## Results

### Cross-modal audiovisual task

We trained mice on a cross-modal audiovisual go/no-go task (Fig. 1a–c). The task consisted of four blocks: two unimodal sensory discrimination blocks where animals licked or refrained from licking either based on the orientation of the visual stimulus (45° or 135°) or the frequency of an auditory tone (5 kHz or 10 kHz) and two cross-modal discrimination blocks where animals were presented with both auditory and visual stimuli and made their decision to lick based on one modality while ignoring the other (Fig. 1a). The unimodal blocks preceded the cross-modal block in which the modality that was relevant for the decision was the same as the modality of the unimodal

block (Fig. 1c). The modality of the initial unimodal block was randomized across animals. Critically, during the cross-modal blocks, the two stimulus modalities were presented simultaneously but the relevant modality was uncued and it was the task context that determined the relevance of a particular modality with respect to the choice that the animals had to make. Only licking or refraining from licking during the final second of the 3-second visual stimulus was considered as the animal's decision (Fig. 1b). Misses and false alarms led to a timeout. The stimulus modality in the initial block was selected randomly for any given day and the modality of the subsequent block, the transition block, was the other modality.

We trained 8 animals to perform the context-dependent decision making task (Fig. 1d). Mice were first trained on the visual discrimination task over multiple sessions (eight sessions on average, Fig. 1d, light blue, see "Methods" for details); next, they were trained on the auditory discrimination task over the subsequent sessions (five sessions on average, Fig. 1d, light green); and finally on the attend-visual and attend-auditory cross-modal tasks in random order within a session over 15 training sessions on average (Fig. 1d, dark gray). Animals performed 300–500 trials daily. Performance of individual animals was measured using the d' statistics, which measures the standard deviation from chance performance during 'lick' and 'no-lick' trials (chance d' = 0, see "Methods"). Training of the animals was considered complete when their performance in the cross-modal task exceeded d' = 1.7 (probability of chance behavior <0.1%). All animals showed average d' values above the threshold in all behavioral tasks (Fig. 1d).

We used detailed behavioral modeling in order to reliably correlate potential changes in behavioral strategies with neural signatures of task performance. Trials were distinguished based on three factors: 1, Task context; 2, Congruence of multimodal stimuli, i.e. whether the auditory and visual stimuli indicated the same action in the two contexts; 3, Task outcome, i.e. whether the expected behavioral response was 'go' or 'no-go'. Animals tended to commit errors more frequently in 'no-go' trials and in conflict trials, which required context inference and active conflict resolution (Fig. 1e). Poor average performance in conflict 'no-go' trials prompted a detailed analysis of performance: low performance can be both a result of a constant high error rate or shifts between behavioral strategies. We analyzed trial-by-trial performance to identify periods where task-consistent behavior was present (Fig. 1f). Binary responses prevent strict trial-by-trial analysis of the behavior. Instead, we assumed across-trial constancy of a strategy and therefore we calculated a running average of choices in different trial types. Consistent blocks are identified with blocks of trials where all four trial types (congruent/incongruent, and 'go'/'no-go', Fig. 1f) showed above chance performance. The moving average spanned 20 trials, ensuring both continuity of consistent blocks and a sampling of all trial types. The identified consistent blocks could not result from making random trials with a lick frequency matched to the measured tendency of the animals to lick (Fig. 1f inset, "Methods"). Consistent blocks of trials in both the attend-audio and attend-visual contexts were found in all animals (n = 8, Fig. 1g). No systematic difference was found in the length of consistent blocks between the two contexts across animals, and half of them were characterized with balanced lengths. In these consistent blocks performance of the animals was balanced across contexts and, importantly, successful 'go' and 'no-go' trials were also balanced (Fig. 1h).

To directly test if contextual modulation of choices underlies behavior, we formulated alternative behavioral models corresponding to behavioral strategies with and without contextual effects. We compared alternative models based on their log likelihood for the trials in consistent blocks. First, we compared a context-aware model to a model that based its choices on the opposite, contextually irrelevant modality. Then we compared an extended version of the context aware model that featured nonspecific noise variables to account for biases and lapses to the context unaware model that ignored the

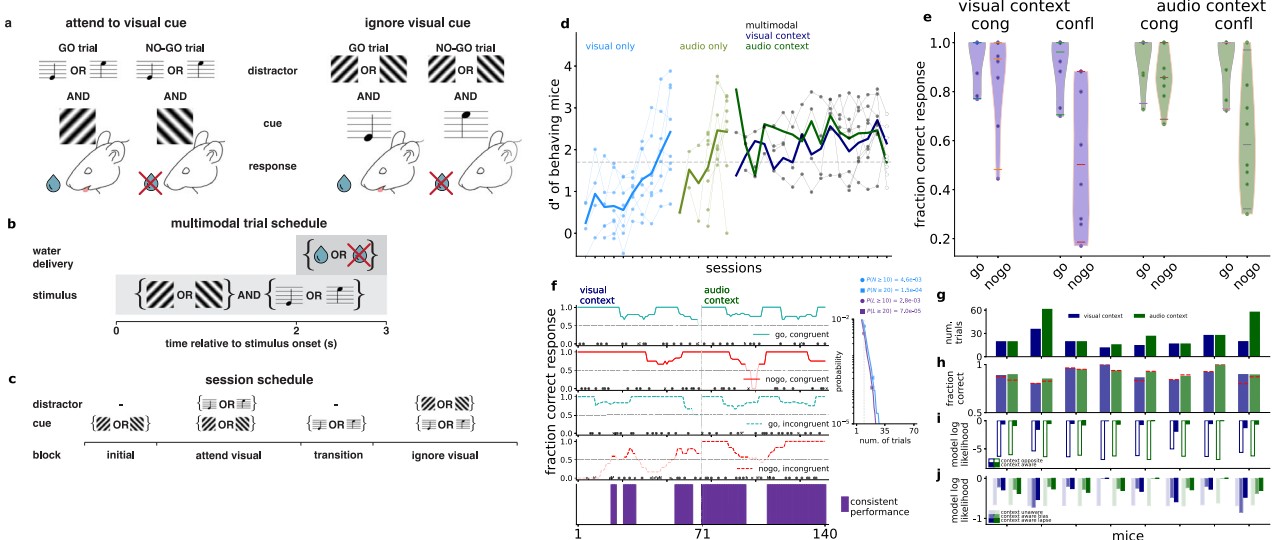

**Fig. 1 | Behavioral paradigm. a** Stimulus and reward structure in the 'attend visual stimulus' (*left*) and 'ignore visual stimulus' (*right*) tasks. **b** Trial structure. Stimuli were presented for three seconds, and a three-seconds rest period followed (time-out extended the delay to ten seconds). Water was available after two seconds from start. **c** Structure of an experimental session. **d** d' values for individual sessions of 8 animals, in the visual and auditory discrimination tasks (light blue and light green, respectively), followed by the cross-modal task, where each session combines the attend visual (dark blue) and attend auditory (dark green) context in random order; expert level performance was defined when combined d' exceeded 1.7 (dashed horizontal line). Each filled dot represents the averaged d' of one animal in a given training session, the empty circle represents the recording session. Individual animals (thin lines) and averages over mice (thick lines) are shown. Most mice were trained for fewer sessions than the full duration of the three session types. Session days are aligned to the last session for each animal. Multimodal sessions are overlaid for easier comparison of performances. **e** Fraction correct of responses grouped by congruence and expected action ('go' or 'no-go') in the two contexts (*blue* and *green*). Individual animals (*n* = 8, dots) and the whole population (violin plots, red line: mean across animals) are shown. **f** Behavioral performance for an example animal during a recording session for different trial types (top four

panels). Success and failure trials are marked by filled circles and crosses, respectively. Lines show moving averages, and darker shading indicates above chance performance. Consistent trials (bottom panel, *purple*) correspond to periods with above chance performance on all four trial types. *Inset*: probability of (*horizontal axis*) the number of trials, *N*, in (*light blue*) and length of consecutive consistent trials occurring at least once, *L*, of (*purple*) the session in one context (70 trials) under a model in which the animal makes random decisions with lick rate matching the empirical rate in incongruent trials (*p* = 0.75). Legend: cumulative probability of at least (≥) the number and at least one occurrence of at least the length for the criteria for consistency (*N,L* = 10, *vertical dashed line, circles*) and typical mice (*N,L* = 20, *squares*). **g** Number of consistent trials for individual mice (*n* = 8) for the visual (*blue*) and audio (*green*) context. **h** Fraction correct performance for all trials in consistent periods (*colors as above*), and for incongruent 'no-go' trials only (*red dashed lines*). **i**, Model log-likelihoods averaged over consistent incongruent trials in the visual (*blue*) and audio (*green*) context for individual mice (*n* = 8), for a model targeting the opposite modality (*empty bars*), and the correct modality. **j** Same as *i*, but with a context agnostic model with mean choice lick bias (*faint colors*), and a context-aware model with a fitted bias or lapse parameters (intense *saturated colors*).

stimuli completely but featured a bias term which determined the tendency of the animal to lick in a given trial. These two comparisons addressed the two possible cases of non-performance: opposite modality interference and random choice. Since incongruent trials were most challenging for the animal, we constrained our analysis to incongruent trials and evaluated the alternative models based on the likelihood of the choices of individual animals. In the first comparison, the context aware model fared better than the opposite irrelevant modality based model (Fig. 1i). In the second comparison for biases and lapses, the context-aware behavioral model consistently had the highest likelihood for the choices each animal made at either a reasonable bias ($\beta$) or lapse ($\lambda$) parameter ($p_{go}$-1+ $\beta$, $p_{nogo}$ - $\beta$, and $p_{go}$ ~ 1 − $\lambda$, $p_{nogo}$ ~ $\lambda$ where $\beta$ = 0.08 ± 0.07, and −0.01 ± 0.05, $\lambda$ = 0.13 ± 0.03, and 0.07 ± 0.02 over mice for visual and audio contexts, respectively, Fig. 1j, "Methods"), confirming that consistent trials were characterized by contextual modulation of behavior.

### Non-visual variable related activity in V1
Neural activity was recorded from all V1 layers on 128 channels (two 64-channel shanks) with extracellular silicon probes[16]. Spiking activity of sorted units was obtained by applying kilosort2[17] followed by strict manual curation. Importantly, as task context is characterized by slow dynamics we paid specific attention to electrode drift as it could introduce long time-scale correlations in the recorded activity, potentially confounding the analysis of context representation. To prevent this, we relied on a conservative spike sorting approach. Units

were tested individually for drift by assessing the constancy of signal to noise ratios on amplitudes, feature projections, and spiking frequency across the recording session (see "Methods" and Supplementary Fig. 1) and were discarded from further analysis if showing signatures of drift even on a short part of the session. The stringent selection criteria yielded 196 units across all mice (see Supplementary Fig. 4 for the distribution of units across mice).

Visual stimulus presentation to the monocular contralateral visual field induced substantial modulation of the activity of V1 neurons (mean firing rate changed from 6.79 ± 0.66 to 8.14 ± 0.74 Hz from baseline to stimulus presentation, trial-to-trial variance on stimulus, mean and s.e.m. over neurons: 22.92 ± 2.95 Hz$^2$, number of neurons 196 from 8 mice) including neurons whose firing rate was increased (5.78 ± 0.63 to 8.56 ± 0.78 Hz and trial to trial variance on stimulus: 26.23 ± 3.31 Hz$^2$, 138 neurons) or decreased (9.19 ± 0.70 to 7.13 ± 0.62 Hz and trial to trial variance on stimulus: 15.83 ± 1.71 Hz$^2$, 58 neurons), respectively (Fig. 2b). Mean waveform trough-to-peak time and amplitude clustering revealed 73 narrow spiking and 123 broad spiking units, with firing rate change to stimulus from 10.40 ± 1.50 to 12.06 ± 1.63 Hz and 4.65 ± 0.47 to 5.81 ± 0.58 Hz, respectively. Individual neurons showed sensitivity, quantified as mean response difference in units of standard deviation (see "Methods") to visual stimulus identity, auditory stimulus identity, or the choice the animal was about to make (Fig. 2b). In addition, we also found neurons that displayed distinct activity levels in the two task contexts (Fig. 2b). Individual neurons showed mixed selectivity to multiple variables.

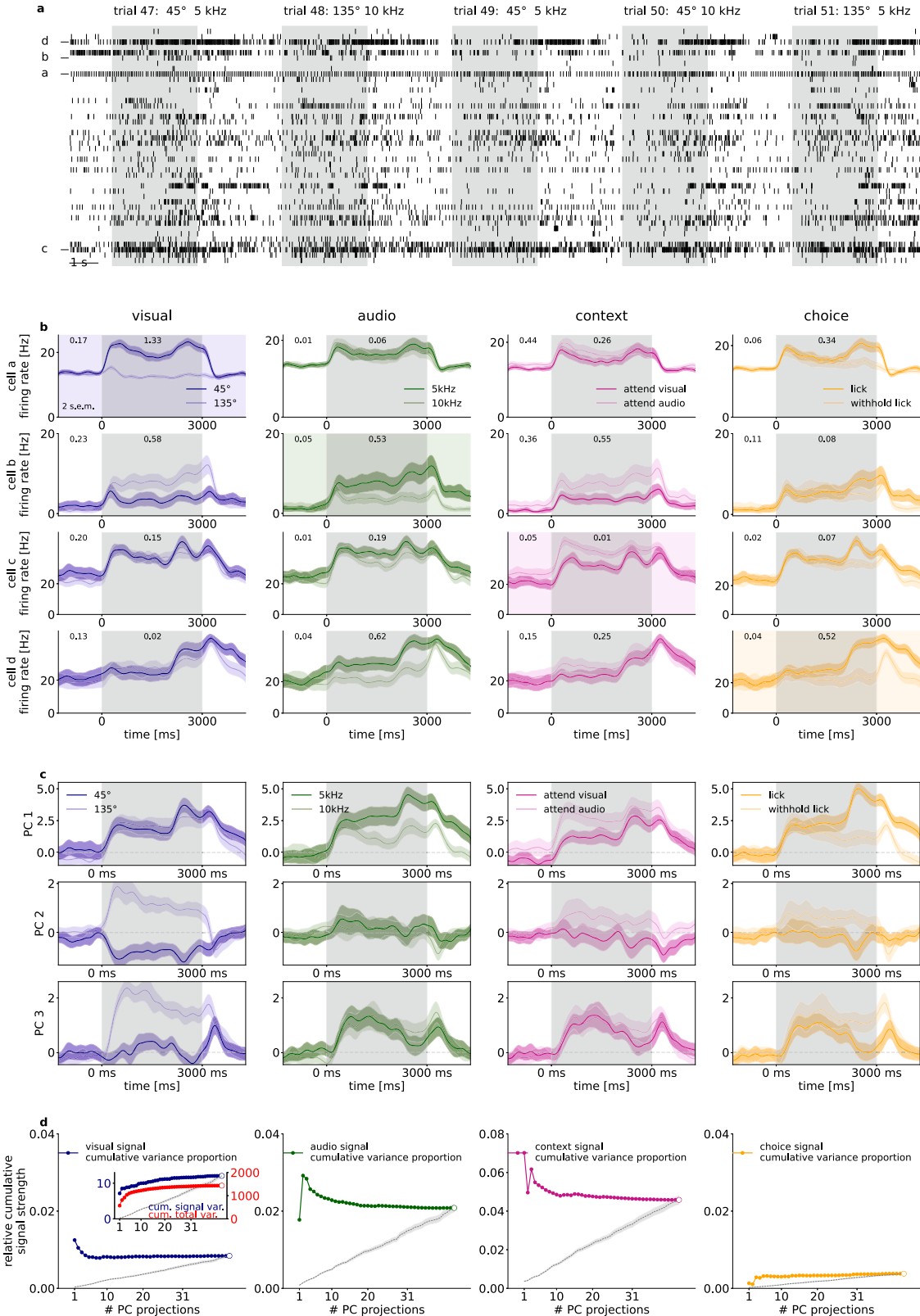

Selectivity of individual neurons showed that all relevant task variables were represented in V1 but it remains unclear how dominant this is in the population activity. We measured the task-variable elicited variance, the variance in mean responses to the two distinct values of a particular task variable. The relative task-variable elicited variance was obtained by normalizing with the total variance in population activity, resulting in 1.89 ± 0.41% (visual), 2.66 ± 0.62% (audio), 3.16 ± 0.48%

(context), 1.24 ± 0.25% (choice), averaged over n = 8 animals (Fig. 2d, open circles). The dominance of these task-related variances in population activity was assessed by measuring task-variable contributions to the principal components (PCs) of population activity, termed signal variance. Task variables could be readily identified in individual PCs, indicating a prominent contribution to population activity (Fig. 2c). We calculated relative contributions of individual task

**Fig. 2 | Population recording from the V1 of an example mouse. a** Spike raster for the recorded population of 31 neurons in five subsequent multimodal trials in an example animal. Both stimulus presentation period (gray background) and inter-trial intervals (white background) are shown. Letters labels indicate neurons on (**b**). **b** Sensitivity of four example neurons (*rows*) to different task-relevant variables (*columns*). Across-trial average firing rates for two values of the particular variable (*dark and light lines, shadings* around lines denote 2 s.e.m., values are indicated in the legend). Sensitivity to a particular task variable (difference in means normalized by the standard deviation and averaged across time points) are shown both for the stimulus presentation period and for the intertrial interval (numbers at the top of panels). **c** Task-relevant modulation of population activity as shown by Principal Component Analysis (PCA) performed on the stimulus presentation period. Time

course of population activity projected on the first three principal components (*three rows*) averaged across different values of the four task-relevant variables (*four columns*). Colors are matched with those on (**b**). **d** Relative signal variance of population activity associated with the four different task-relevant variables (*four columns*). Variance is normalized by the total variance of the population responses. Relative variances are shown cumulatively for activities projected on a growing subset of PCs. Inset shows signal and total variances on separate vertical scales, the numerator (*dark blue*), and total variance (*red*), the denominator, of the relative variance shown in the main panel. Empty circles correspond to the total variance. A baseline relative variance is established by using a random orthogonal projection matrix (mean, and s.e.m. of 20 random projections, *gray dashed lines and bands, respectively*).

variables to each PC. For this, we first assessed the signal variance in individual PCs and then the cumulative signal variance was established as the sum of variance explained by the specific task variable in a subset of PCs (Fig. 2d, inset, see also "Methods" for details). Finally, we calculated the relative contribution of a specific task variable to the total variance in the population as the ratio of cumulative signal variance and total variance captured by the same subset of PCs. The relative cumulative signal variances peaked at low dimensional PC subspaces, indicating that all task variables are prominent in low dimensional subspaces (Fig. 2d). A random orthonormal basis resulted in markedly lower relative cumulative variance. In summary, population activity in V1 is influenced by all relevant task variables and these task variables contribute to the leading directions of variance.

**Orthogonal representation for sensory stimuli and task context**

The above analyses highlighted that the task context is represented in the population activity of V1 neurons. To investigate how V1 differentially represents identical stimuli when these stimuli indicate different behavioral outcomes, we constructed linear decoders (Fig. 3a) for visual stimulus identity (Fig. 3b) and context (Fig. 3e). We fitted separate decoders in 50-ms time windows (Fig. 3b, e, for details see "Methods"). To distinguish decodability from chance, we compared decoder accuracies to randomized label decoders (shuffled baseline, see "Methods"). Both visual stimulus and task context could be decoded throughout the stimulus presentation period, and this was consistent across animals (Fig. 3c, f, mean visual accuracy $0.70 \pm 0.06$, 2 s.e.m. of visual accuracy $0.03 \pm 0.00$, mean context accuracy $0.67 \pm 0.04$, 2 s.e.m. $0.03 \pm 0.00$, number of animals: 8). We assessed the overlap between visual and context representations by comparing the contribution of individual neurons to the two linear decoders (Fig. 3d, g). We found strong overlap between the weights of visual and context decoders, confirming mixed selectivity for stimulus and context. We then asked if the activity of narrow-spiking, putative interneurons, could potentially encode context. When constraining our analysis to narrow spiking neurons, we found stable context representation (Supplementary Fig. 2c, d). The analysis was repeated for only broad spiking cells. We found that accuracy of decoding context from broad or narrow spiking neurons was close to that from all neurons (Supplementary Fig. 2) indicating that context is not represented solely in narrow or broad spiking neurons.

We investigated whether stronger neural context representation corresponds to better behavioral performance. We used the trained context decoders to assess population activity separately in trials when the behavior indicates consistent context-aware choices (consistent blocks) and in trials outside this regime ('exploratory blocks', Figs. 1f, 3h). The number of trials between contexts and performance levels were equalized by repeated subsampling (Fig. 3i, see "Methods'). Decoder performance differences between consistent and exploratory trials were significantly positive in most animals ($n = 6$ out of 8) and also when using the distribution of differences from all animals (Fig. 3j). Importantly, enhanced contextual modulation of population responses in consistent blocks indicates that behavior that is more

consistent with task rules is actually correlating with stronger contextual modulation.

A context representation that is not independent from visual content representation could yield correlations that are detrimental to the visual stimulus representation. We therefore investigated the relationship between the visual and context representations by analyzing the properties of their decoders. Using our Multidecoder Subspace Analysis framework (MDSA, see "Methods"), we constructed a trial-by-trial measure of population activity: population vectors constructed from spike counts during the first 1.5 s after stimulus onset were projected on a two-dimensional subspace defined by the DVs (Fig. 3a) of the two investigated variables (Fig. 3k). MDSA for the visual and context decoders revealed a striking pattern (Fig. 3l): population responses to different task contexts and different visual content formed distinct clusters, and variations across contexts and stimulus identities were close to orthogonal. Orthogonality was consistent across animals (Fig. 3m), highlighting that visual stimulus and context were represented in independent subspaces albeit with overlapping populations (Fig. 3l). Time-resolved analysis of the DV angle showed consistent orthogonality during stimulus presentation (Fig. 3n, "Methods"). We constructed two additional tests of the geometry of context and visual representations. First, we investigated context-related activity along the dimension in which the visual stimulus was represented. Decoding context from neural activity along the visual DV yields close to chance performance (Supplementary Fig. 3a, b). Second, we analyzed if augmenting the one-dimensional space identified by the context decoder by the visual DV results in improvement in context decoding, a signature of not independent representations. We applied an outer two-fold cross-validation scheme: we projected activity in test trials onto the context DV basis fitted in training trials. Adding the visual subspace to the context subspace did not increase the context accuracy (Supplementary Fig. 3c, d). Thus, the visual subspace does not contain linearly decodable information about context.

Both the activity of individual neurons and that of the neuron population indicated sensitivity to the auditory modality (Fig. 2). In order to establish if the representation of auditory stimulus is organized along similar principles as that of visual stimulus, we performed the decoding analyses for auditory stimuli too. Auditory stimuli could be decoded from population activity (Supplementary Fig. 5a, b) and contribution to auditory decoding was distributed across the recorded population of V1 neurons (Supplementary Fig. 5c). Neurons contributing to the auditory decoder did not show consistent correlations with those contributing to context decoding (Supplementary Fig. 5d). MSDA revealed that the decoder for the auditory stimulus was orthogonal to the decoder learned for task context (Supplementary Fig. 5e–g). Thus, the population activity in V1 is organized such that beyond the visual modality, the other task-relevant modality is also represented, and variance elicited by sensory variables are orthogonal to the context representation.

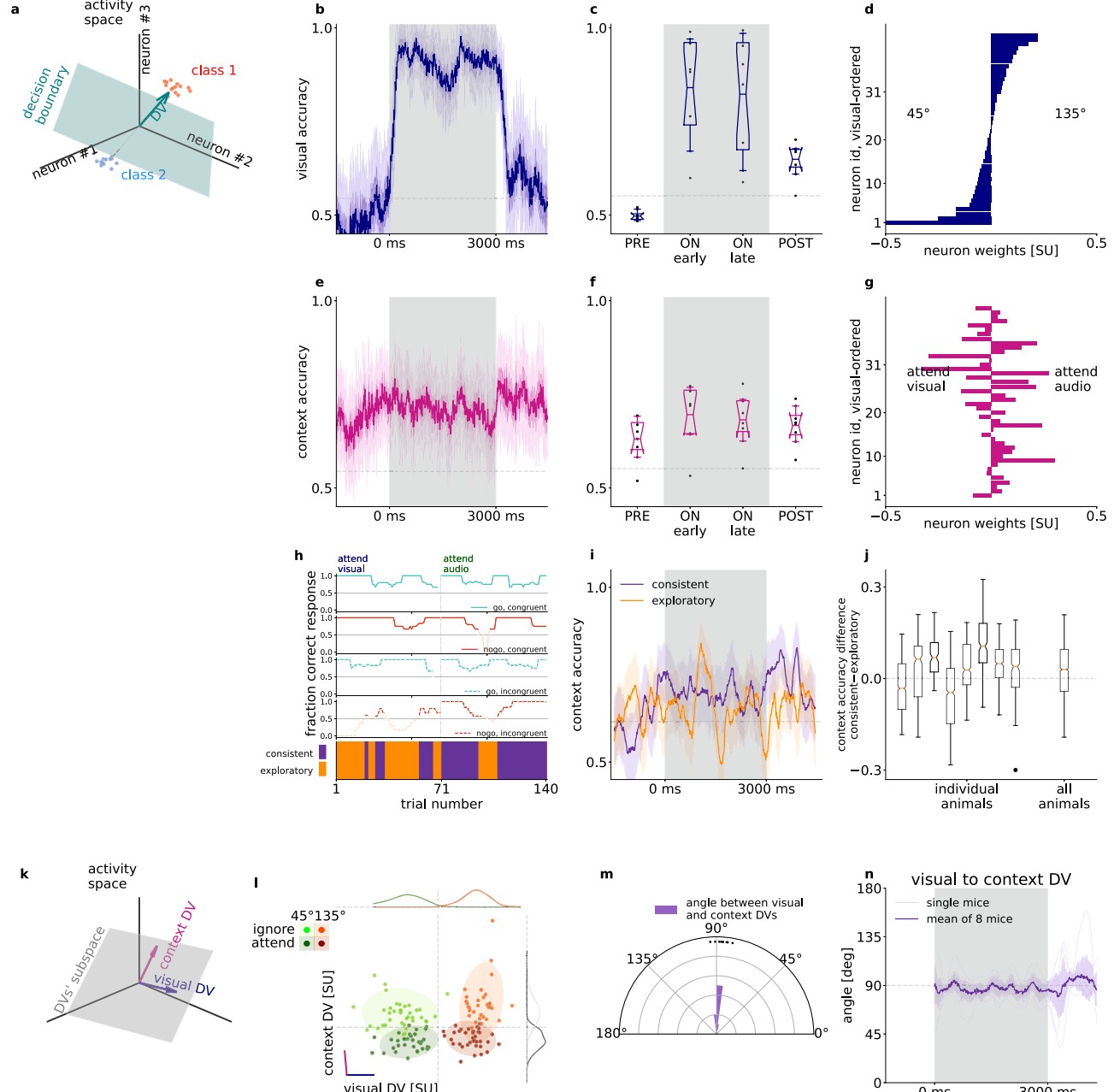

**Fig. 3 | Representation of task context in population activity. a** Illustration of the decision vector of a linear decoder constructed for neural activity. A linear decoder discovers a decision boundary (green surface) in the activity space which separates responses in individual trials (*dots*) in two conditions (*red* and *blue*). The decision boundary can be uniquely characterized by the decision boundary normal vector (decision vector, DV, *green arrow*). **b** Visual decoder performance in 50 ms time windows at 10 ms sliding resolution for an example animal. Gray background represents the stimulus presentation period. Gray dashed line indicates shuffled baseline. Note that firing rates are estimated by using convolution kernels with 100-ms characteristic width and therefore decoders contain information from future time points, resulting in a slight increase in decoder performance prior to stimulus onset. **c** Average performance of visual decoder for all animals (*dots*, $n = 8$), prior to stimulus onset (PRE), during stimulus (ON), and after stimulus (POST). Box and whiskers denote 25–75, and 2.5–97.5 percentiles respectively, midlines are mean, notches are 95% confidence level error of the mean. Gray dashed line indicates shuffled baseline. **d** Contributions of individual neurons to the visual decoder (decoder weights) arranged according to the magnitude of the weights. **e-g** same as (**b-d**) but for decoding context from the population activity. Ordering of neurons on **g** is the same as that on (**d**). **h** Behavioral performance in 'go' or 'no-go', and congruent or incongruent trials (top four panels) for an example animal. Darker

color indicates behavioral performance exceeding chance. Bottom panel: Consistent (*purple*) and exploratory trials (*orange*). **i** Context decoders, as in (**e**), using only consistent (*purple*) or exploratory (*orange*) trials from both contexts; s.e.m. of leave one out mean accuracy (*bands*), effective chance level (*gray dashed line*), same animal as (**h**). **j** Distribution of time-resolved accuracy difference between consistent and exploratory accuracy, excluding timepoints where both consistent and exploratory accuracies are below chance for individual animals (*left*), and for all animals concatenated (*right*); each animal has $n_T > 450$ time points for the distribution, boxplot parameters as in (**c**). **k** Combination of multiple DV bases forms a new basis that defines a higher dimensional subspace of task relevant population activity. **l** Population responses of an example animal, averaged over the first 1.5 s of stimulus presentation projected on the DV subspace in individual trials (*dots*), and their estimated normal distribution (mean and 2 std, *shaded ovals*) in different task contexts (*dark* and *light*) and with different visual stimuli presented (*red* and *green*). Purple and blue lines denote the DV directions of context and visual decoders, respectively. Histograms show population responses projected on orthogonal components of single DVs. **m** Histogram of the angle between context and visual DVs across animals (*dots*). **n** Time course of the angle between the visual and context decoders throughout the trial, distribution of animals ($n = 8$, *gray faint lines*) mean (*purple thick line*) and s.e.m. (*faint purple band*).

## Context representation in the absence of stimulus

Task context is not independent across trials since context is invariant in a given block, therefore context needs to be maintained during the intertrial intervals. Time-resolved decoding of context demonstrated persistent context-related activity during pre-stimulus and post-stimulus activities with similar decoder accuracies (Fig. 3b, pre-stimulus mean accuracy across animals $0.63 \pm 0.04$, during stimulus mean $0.69 \pm 0.05$, post-stimulus mean $0.67 \pm 0.03$). Context decoder accuracy varied across animals but decoding accuracy before and during stimulus presentation were strongly correlated (Fig. 4a, $R = 0.82$, $p = 0.002$). By controlling for the number of units in the analysis, we confirmed that this dependency was not solely dependent on the number of neurons recorded (Supplementary Fig. 4).

Despite the constant presence of the context signal throughout the trial, it remained uncertain whether the context representation during pre-stimulus and on-stimulus activities were invariant or underwent dynamical transformations. We investigated this question by time-shifted decoding: we constructed decoders at reference time points and tested them at different time points during the trial (Fig. 4b). As a reference, we performed the analysis on visual signals. Time-shifted visual decoder performance was relatively stable even with long time shifts. Time-shifted decoding for all training and testing time points revealed a persistent and stable representation throughout stimulus presentation (Fig. 4c). Stability of the representation was quantified by calculating the slope of the linear fit on the time-shift-dependent decoding performance (Fig. 4c inset). As expected, the decay rate was low during stimulus presentation (Fig. 4d). Consistency of the representation was also assessed by calculating the angle between DVs of decoders trained at different times (Fig. 4e). DV angle dropped from close to 0 to about 45 degrees with small time differences but remained stable throughout the stimulus presentation.

To assess the stability of the context representation throughout the trial, we performed time-shifted decoding analysis. The decay of the time-shifted accuracy appeared to be faster for task context than for the visual content (Fig. 4d, g, 8 mice, average decay rate during stimulus, dependent one-sided t-test $p = 0.05$). Importantly, during both the pre- and on stimulus periods the time-shifted decoding showed a relatively stable representation of task context (−1500 to −500 and 1000 to 2000 ms average decay rate of $0.26 \pm 0.05$ and $0.26 \pm 0.05$ accuracy loss/100 ms, respectively, $n = 8$ mice), but stimulus onset was characterized by an abrupt decay (−200 to 200 ms average decay rate of $0.45 \pm 0.09$ accuracy loss/100 ms, 8 mice, Fig. 4g). This finding was corroborated when assessing larger than 100 ms time-shifted DV accuracies (Fig. 4f, example animal) and angles (Fig. 4h, example animal) during the whole pre- (Fig. 4f inset, yellow, bottom left), and on-stimulus (Fig. 4f inset, yellow, top right) periods, as well as across their borders (Fig. 4f inset, red): There were higher accuracies and smaller angles both within the pre-stimulus ($0.64 \pm 0.03$, $\sigma = 0.09$ and $51.9 \pm 4.8°$, $\sigma = 13.5°$) and within the on-stimulus periods ($0.65 \pm 0.03$, $\sigma = 0.09$ and $49.1 \pm 4.7°$, $\sigma = 13.2°$), but low accuracies and closer to orthogonal angles across the border of pre- and on-stimulus activities ($0.57 \pm 0.03$, $\sigma = 0.08$ and $76.4 \pm 5.7°$, $\sigma = 16.2°$, all mice, $n = 8$). Such drop in across-time decoding performance and increase in decoder angles indicated that the neuronal population activity underwent a transformation upon stimulus onset. This corresponded to a change in modulation of individual neurons (Fig. 4i, j). Such transformation was characteristic to most animals (Fig. 4f, example animal) but some animals displayed more stable context representation (Fig. 4k, second example animal). To assess if the persistence of context representation in these animals can be characterized by the one-dimensional space defined by our linear context decoder, we investigated the nullspace of the context DV, i.e. the activity subspace which the direction of the decision vector was subtracted from (Fig. 4l, see "Methods"). We found that in these animals

context could be decoded from the nullspace and across-time decoding displayed abrupt changes at stimulus onset and stimulus offset (Fig. 4l). This suggests that the population consists of neurons that persistently represent context (Fig. 4i, j), but a large subpopulation of neurons undergoes a transformation upon stimulus onset (Fig. 4i, j). Defining the lack of stability to stimulus perturbation, "blockiness", as the difference in accuracy across the pre- and on-stimulus borders (Fig. 4f, see "Methods"), we found that blockiness increases significantly when decoding from the nullspace of the context decoder compared to the full neural space ($p = 0.029$, two-sided paired t-test, $n = 5$ animals, Fig. 4n). Note, that we only included animals whose mean off-diagonal pre and on stimulus (Fig. 4f inset, yellow), or mean nullspace-decoded accuracies were above shuffle control. In summary, while task context is represented throughout the trial in V1, stimulus onset perturbs the context representation observed during the pre-stimulus period.

## Context invariant visual responses

Contextual modulation of V1 responses and stimulus-induced responses were shown to affect overlapping neuron populations (Fig. 3d, g). It is therefore critical to assess if the contextual signal interferes with the sensory signal. We address this issue by investigating how the exact same stimulus affects the population under different contexts.

We investigated the temporal evolution of population activity in the subspace defined by the visual and context decoders and calculated the trial-averaged neural responses to either 45 or 135 degree gratings in the two contexts for correct trials (Fig. 5b). Change in context introduced a shift in the trajectory but left the time course of the response largely intact (Fig. 5b–d). In the visual subspace, the across-animal contextual difference of population responses was relatively small ($0.30 \pm 0.003$) compared to the projected absolute standardized mean rates (mean $1.01 \pm 0.008$, std 0.76, $n = 8$ animals × 300 timepoints), and varied over timepoints within a small portion of the distribution of differences (std 0.23, Fig. 5e, g). In contrast, in the context subspace typical projection rates were smaller (mean $0.55 \pm 0.004$, std 0.40, $n = 8$ animals × 300 timepoints), while the mean difference ($0.71 \pm 0.006$) was larger than in the visual subspace ($p < 0.05$ at all timepoints between 500–2500 ms after stimulus onset, two-sided two-sample t-tests for each timepoint and both of 45 and 135 gratings), while it had larger variance (std 0.24 and 0.42, Fig. 5f, h). To quantify the invariance of visual representational subspace, we constructed separate visual decoders in the two contexts. We characterized individual animals by the difference of time-averaged visual decoders from the first 500 ms of stimulus presentation (Fig. 5a). We limited averaging to the first 0.5 s of stimulus presentation to avoid interference with choice-related activity. Although the mean accuracy difference varied across animals, their average was not distinguishable from 0 (two-sided t-test, $p = 0.24$). Thus, a change in task context shifts the entire visual response trajectory in a subspace orthogonal to the one encoding visual information, while leaving the visual dimension invariant across contexts.

We repeated the analysis for the auditory modality, and found similar results (two-sided t-test, $p = 0.65$, Supplementary Fig. 5h). Although V1 is not the primary area of low-level audio information processing, our results reveal a consistent representational scheme for visual and auditory variables.

## Choice- and context-related activity in V1 are orthogonal

Both individual neurons and population responses showed sensitivity to the choice animals were about to make. Choice-related activity was not expected to be independent from activity related to stimulus identity since the choice depended on it. Indeed, projecting neural activity onto visual-choice or audio-choice subspaces in each context showed that hit and false alarm choices correlated with 'go' signals, as

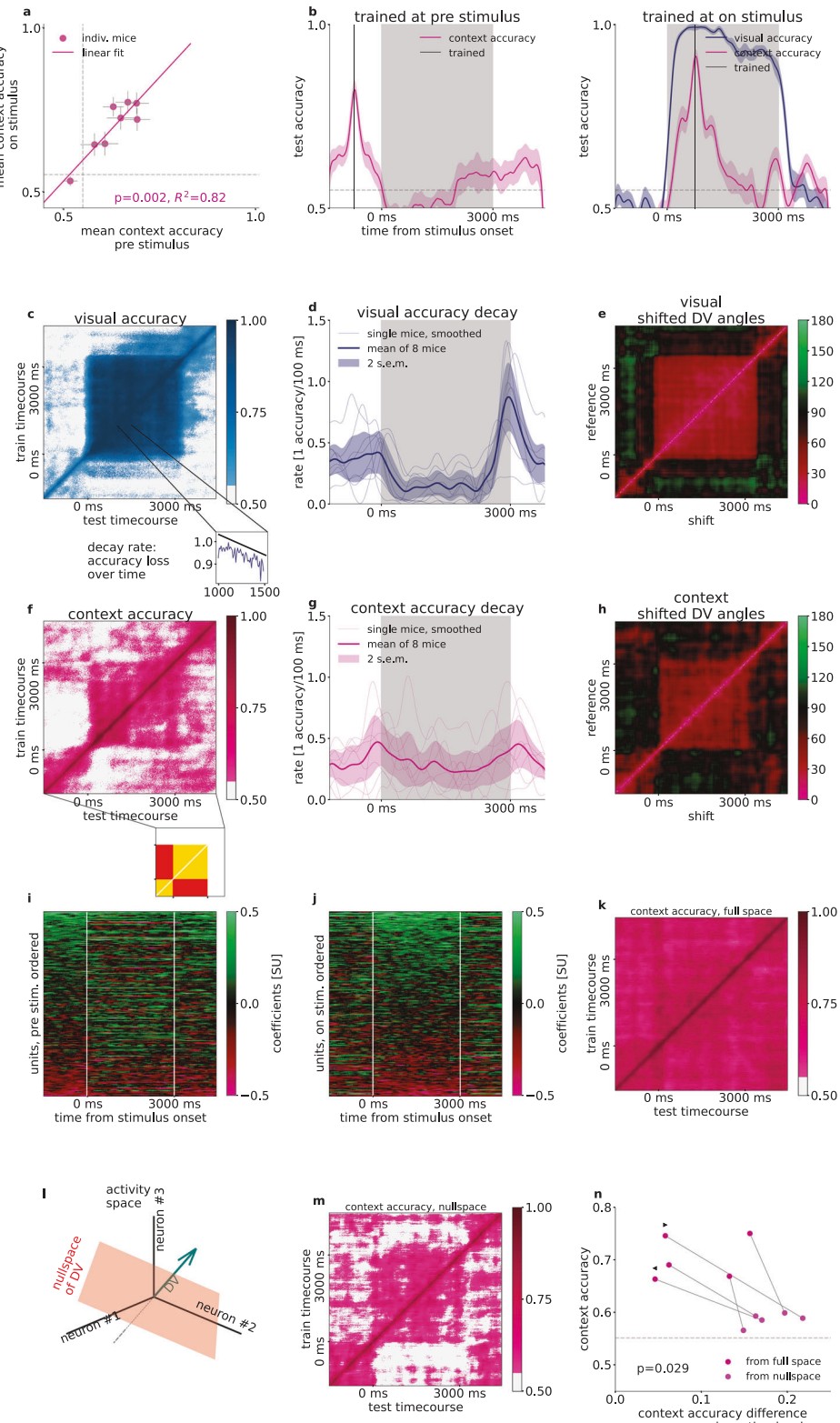

well as correct rejections and misses correlated with 'no-go' signals, thus neural response patterns appeared in a skewed geometry (Supplementary Fig. 6a, b, example animal). We found that the angles between DVs for stimuli and choice were 57.52 ± 6.29° in the visual context, and 65.47 ± 5.67° for the auditory context, significantly below 90° (one-sided t-test, $p_{visual}$ = 0.001, $p_{audio}$ = 0.002, Supplementary Fig. 6c, d, distribution from all animals). We performed an additional control by testing the independence of visual and audio

representations, as would be expected based on their presentation statistics. Indeed, their representations were orthogonal to each other, evidenced by the angle between the visual and audio DVs derived from the first 500 ms of all trials (91.06 ± 3.13° over mice, not differentiable from 90°, t-test $p$ = 0.76).

To investigate the relationship of choice-related activity to the representation of context, we constructed a linear decoder for choice (Fig. 6a). Choice could be decoded throughout the trial and also after

**Fig. 4 | Stability of context representation. a** Relationship of context decoder accuracies in the intertrial interval and during stimulus presentation. Gray dashed line indicates shuffled baseline. Line, $R^2$ and $p$ values indicate Pearson-correlation, two-sided Wald-test t-statistics ($n = 8$ animals). **b** Accuracy of time-shifted visual (*blue*) and context (*magenta*) decoders. Decoder was trained at a reference time point (gray vertical line) at pre-stimulus (*left*) and on-stimulus (*right*) periods. Gray dashed line as in (**a**). **c** Generalization of the visual decoder across time. Performance (color scale) of decoders trained (*vertical axis*) and tested (*horizontal axis*) at different time windows. *White shading*, decoder performance under shuffled baseline. *Inset*, a cross section of the main panel. *Black line*, linear fit characterizing the rate of performance decay. **d** Smoothed rate of decay at each timepoint in 500-ms windows (thin lines: individual mice, thick line: mean, shading: 2 s.e.m.). **e** Relative angle between the DVs of visual decoders trained at different time windows. **f–h** Same as **c–e** but for the context variable. Inset below (**f**) indicates time-shifted decoder matrix elements for within pre- and on-stimulus periods (yellow) and elements for time-shifts across the border of stimulus onset (red). 'Blockness' = yellow - red. **i** Context decoder coefficients (color code) at each timepoint for all animals ($n = 8$) concatenated. Neurons (vertical axis, $n = 196$) ordered according to mean coefficients during the pre-stimulus period. **j** Same as **i**, but ordered according to the on-stimulus period. **k** Same as **f**, for a second example animal. **l** Schematics of the definition of DV nullspace. **m** Same as **k**, but context is decoded from the context nullspace. **n** Time-averaged diagonal context accuracies from the full neural activity space (*violet*) and from the context nullspace (*light pink*), depending on the blockness in the same subspaces. Dots and lines represent individual animals, $p$-value of two-sided dependent t-test between blockness from nullspace and full space ($n = 5$ for 5 animals, left and right triangles correspond to animals on **f** and **k**, respectively). *Gray dashed line:* Mean accuracies above randomized chance averaged over all animals.

stimulus offset, with higher decoding accuracy towards the end of the stimulus presentation period (Fig. 6b, accuracy early $0.61 \pm 0.02$, late $0.69 \pm 0.03$, one-sided dependent t test $p = 0.016$, 8 animals). Applying MDSA to the choice and context decoders reveals a close to orthogonal relationship between them (Fig. 6c), which was consistent across animals (choice DVs assessed separately in attend: $88.8 \pm 3.4°$, and ignore: $86.8 \pm 2.8°$, Fig. 6d). Multiple sources contribute to choice-related activity, which includes both cognitive factors, such as reward expectation, and movement-related activity, such as preparation of action or locomotion. These factors cannot be distinguished by our analysis but the orthogonality to the context DV demonstrates that choice-related activity comprises a context-independent source of variance in V1.

## Controlling for movement related activity

In our paradigm, task context was stable in the first half of the session and was switched in the second half of the session. As a consequence, changes in movement patterns that occur between these periods can introduce confounds in our analysis of context representation since differences in movement-related activity could be picked up by the task context decoder. To rule out these potential confounds we devised a locomotion matching and a video capture-based motion controls.

We used locomotion data to match the distribution of running speeds across task conditions, similar to firing rate matching[18] and mean matching[19] ("Methods"). Briefly, we constructed a joint distribution of running speeds and population activities in 50-ms windows. Thus, any given time window in a trial yields a point in the multi-dimensional space of running speed and population activity. To reduce sampling noise in this distribution, we collected data from multiple consecutive time windows and data points from these time windows were collectively used to construct a single distribution (Fig. 7a). Separate distributions were constructed for the two task contexts. To control for running speed differences in the two task contexts, we randomly subsampled the joint distribution such that the running speed histograms were matched across contexts (Fig. 7b, c). We found no difference between context decoder accuracies in matched and non-matched trials (Fig. 7d): differences are substantially smaller than the variance yielded by repeated sampling for all animals involved in locomotion matching (Fig. 7e, $n = 3$, average of means and standard deviations $0.0059 \pm 0.0011$, and $0.0212 \pm 0.0011$, respectively). Potential delayed effect of locomotion on V1 activity was investigated by time shifts between the locomotion and neural response data at 100 and 200 ms time windows. We found no effect of this shift on locomotion-matched context decoding performance (Fig. 7f).

Due to the movement asymmetry of 'go' and 'no-go' trials, we tested whether context is still decodable if licking trials are omitted. Note that stimulus statistics had to be identical in the two contexts, thus only congruent trials were used. Using correct 'no-go' trials, there

was no difference between the accuracy of the context decoder compared to the case when all trials were included (Fig. 7g, all animals).

To determine if movement-related activity other than locomotion or licking contributes to decoding context, we performed an additional control by recording video of movement using infrared cameras while recording population activity (see "Methods"). We used two methods to characterize the spatial correlations in video recordings. In the first approach we performed dimensionality reduction on the video stream by applying PCA[12] (see also "Methods"). From the PCA bases we reconstructed the video stream from the first 40 PC time series to assess whether they capture important movements and are free from other noise (Fig. 7h, Supplementary Fig 7i). We tested context representation in trials where movement power was below a threshold derived from the video recordings (Supplementary Fig. 7a). We defined absolute motion as a smoothed average over movement PCs (see "Methods"). Variance over PCs mostly depended on the mean, especially at lower values, reassuring us that we did not incorrectly omit large movements captured by few PCs below the absolute motion stationarity threshold (Supplementary Fig. 7b, c). We allowed stationary trials to have a proportion of timepoints exceeding the threshold (Supplementary Fig. 7d, e, example decoder accuracy time course with fixed threshold = 0.4 and proportion allowed = 0.1, Supplementary Fig. 7a, e, i). We assessed context decoding under a range of thresholds (0.2–2.0) and proportions (0.05–0.50), and showed that stationary or motion trials had no difference in context representation (Fig. 7j). Note that as expected 'no-go' trials are over-represented in stationary trials (Supplementary Fig. 7f). We also tested the context signal at different motion intensities. For this, we calculated motion intensities from the video recordings and defined fixed intervals of motion intensities. We constructed context decoders for different time points during trials such that the number of trials was equalized across contexts. We found that context is decodable from these motion intensity-controlled trials (Fig. 7k, see "Methods"). We found slight differences in movement levels between the two contexts (Supplementary Fig. 7g). Although this effect is eliminated by the equalized levels, we added a second analysis complementing the global PCA method. We identified a number of regions of interest corresponding to body parts of the animal (Supplementary Fig. 8a, see "Methods"), and calculated the mean absolute motion level for each body part from the differential intensities. The distribution of motion levels were either identical or close to identical in the two contexts for most of the body parts (Supplementary Fig. 8b). To eliminate body part movement that can cause different neural patterns between the two contexts, we again decoded context from an equalized number of trials from both contexts available at various motion levels ("Methods"). In this analysis, similar to the PCA based absolute motion level distribution equalization above, only those timepoints and motion levels were used in the averaging, which had at least 5 trials in each context. The detailed multi-body part movement equalization did not have any effect in context decoding from V1 activity, when compared

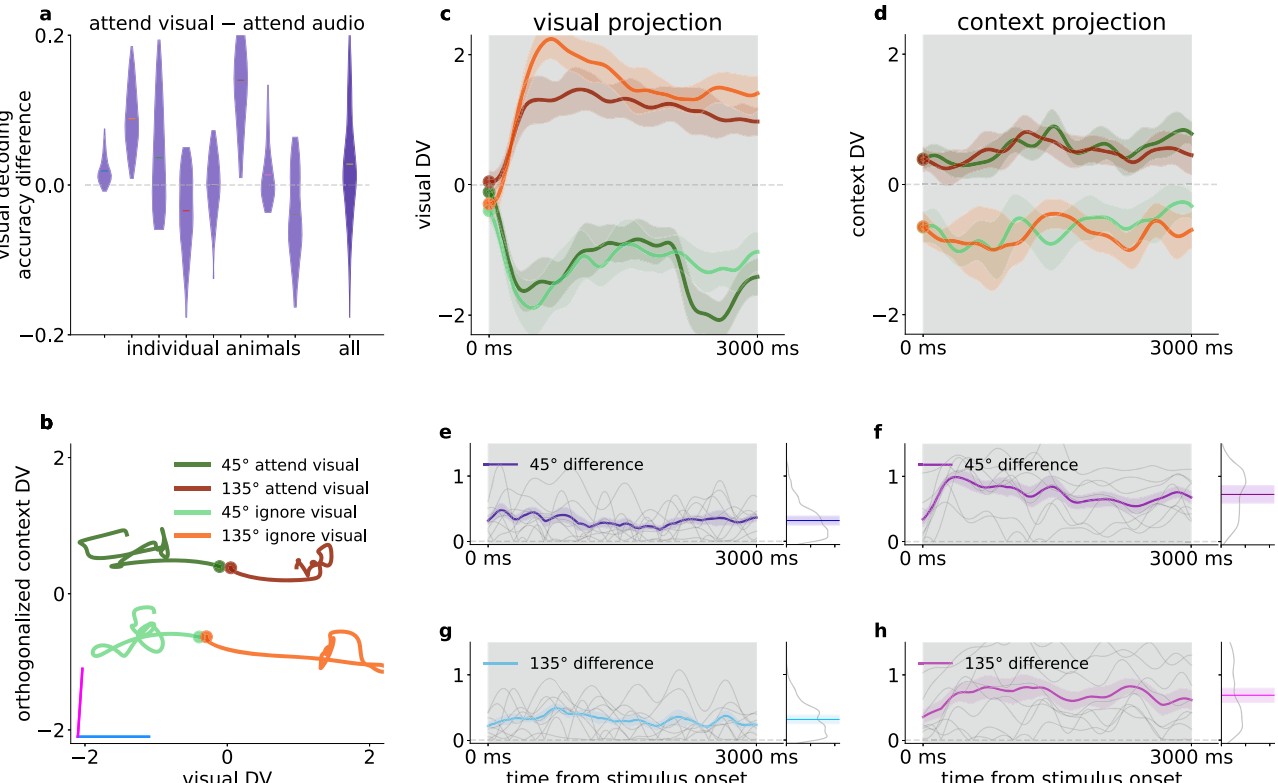

**Fig. 5 | Effect of context on processing visual information. a** Distribution (*violins*) with means (*horizontal lines*) of visual decoder accuracy differences between visual and audio context over each timepoints of the first 500 ms for individual animals (*left ten plots*), and for all mice combined (*rightmost*) **b** Average time course of population trajectory (line: mean, band: 1 s.e.m.) during the on-stimulus period projected on the context and visual DVs for different visual stimuli and for different contexts (*light and dark colors*) for an example animal. *Disks*: stimulus onset. Purple and blue lines indicate context and visual DVs respectively. **c** Average time course of the population response to different stimuli projected on the visual DV for the same animal as in (**b**). Colors as in (**b**). **d** Same as panel **c**, but projected onto the context DV. **e, g** Absolute differences of visual DV-projected population trajectories between different task contexts for the 45° and 135° stimuli, respectively. Individual animals (*thin gray lines*), mean (*thick colored line*) and 1 s.e.m. (*colored bands*) at each timepoint over all animals (*n* = 8). *Right insets*: Histogram of differences from all timepoints and animals (*gray line*), temporal average of the across-animal mean time course (*colored lines*), *shaded bands* correspond to one standard deviation. **f, h** Same as **e, g**, but for context DV-projections.

to an equalized but motion level-shuffled trial randomization (Supplementary Fig. 8c).

In summary, our analyses did not identify any motion-related contributions to the contextual signal in V1.

## Discussion

We showed the existence of a neuronal population representation of a cognitive variable, task context, in V1. Importantly, task context was a latent variable since it was not directly cued and was therefore inferred by the animal through the contingencies between multimodal stimuli and water rewards. Behavioral performance of animals correlated with context representation: blocks of trials in which animals were performing the paradigm well were characterized with better decodability of the context signal. Representation of task context was 'mixed' with that of visual stimuli since overlapping populations showed sensitivity to both. Multi-dimensional subspace analysis revealed orthogonal representation of task context and visual grating stimuli. This multiplexed representation of task variables not only spanned visual and context signals, but extended to responses to auditory signals as well. Furthermore, despite engaging overlapping populations, task context did not affect the population dynamics induced by visual stimulus presentation and was therefore represented independently from visual stimuli. We found a strong signal associated with task context not only during stimulus presentation but also in the inter-trial intervals. While the strength of task context representation during on-stimulus and off-stimulus periods was strongly correlated, the representation underwent a transformation: stimulus onset was characterized by a

dynamical shift in the linear subspace where context-related variability could be identified. In summary, V1 integrates visual inputs with other sensory and cognitive variables by multiplexing these signals, thus ensuring the maintenance of task-relevant variables while avoiding interference with the visual representation.

Flexible use of available information is critical for intelligent behavior[20]. Depending on the context, the same sensory stimulus can evoke vastly different behavioral patterns. Based on behavioral measurements alone, previous studies have proposed that task context is represented through the discovery of latent variables[21–26]. Such latent variable representations were shown to be necessary for efficient learning in multiple cognitive domains[20,24,25,27]. The details of the neural representation of such latent variables have remained elusive. Experiments in both rodents and humans indicate the contribution of the orbitofrontal cortex to the maintenance of latent variables[28,29]. In light of these studies, our results suggest that the effects of the latent task context can be identified as early as the primary visual cortex. Note, that our analyses did not establish a direct link between the observed contextual signal and the rules of the task, therefore the semantics of this signal cannot be unambiguously established. Our findings provide insights into the properties of latent variable representations, yet the way the primary visual cortex contributes to task-dependent processing of sensory signals remains to be established.

Rule or context selective neural firing has been repeatedly observed in frontal and parietal cortices[30–34]. Whether these neural dynamics impact firing in primary sensory cortices during context-dependent decision making is poorly understood. During context-

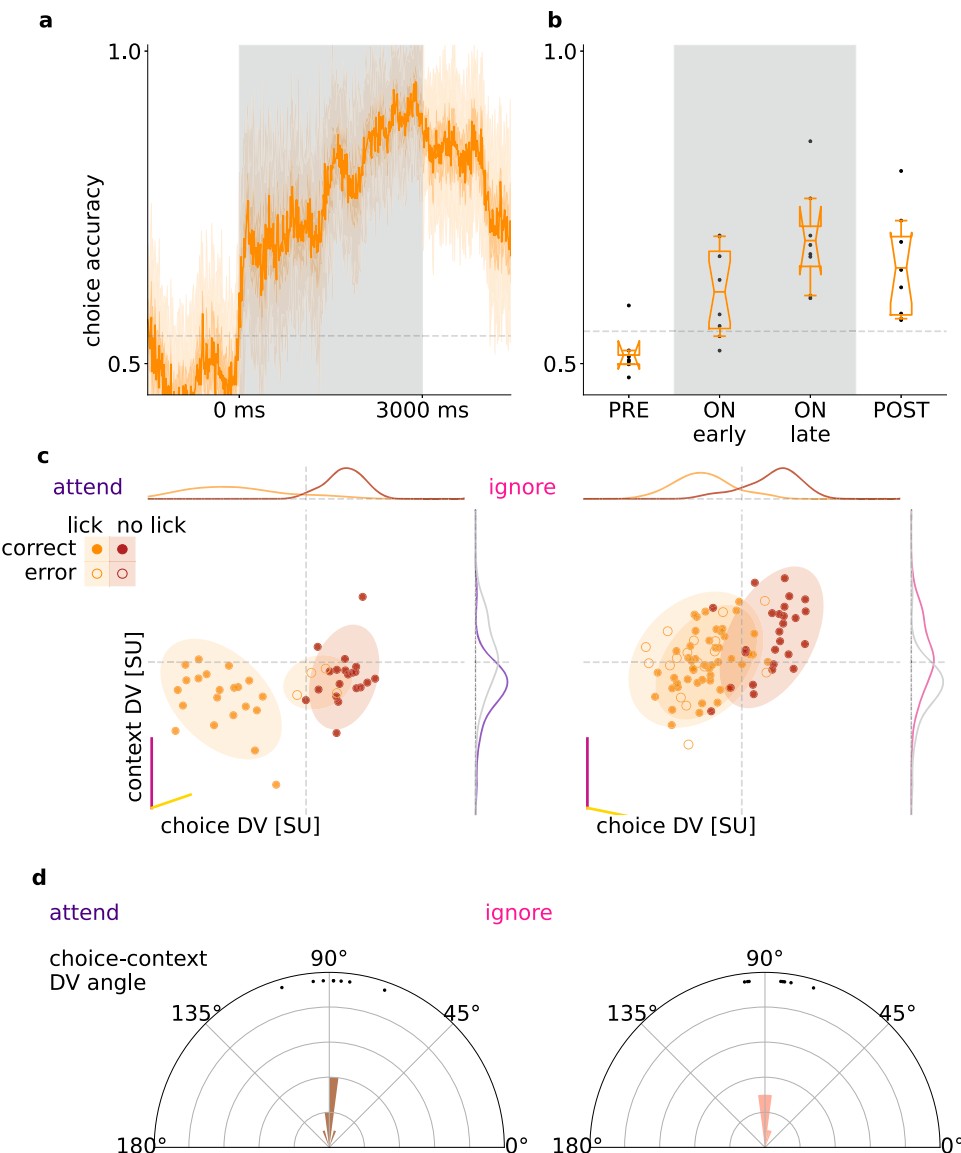

**Fig. 6 | Choice-related activity. a** Decoder performance in 50-ms sliding windows for the choices of an example animal ('lick'/'no-lick'). Gray dashed line indicates shuffled baseline. **b** Average performance of choice decoder prior to stimulus onset (PRE), during stimulus (ON), and after stimulus (POST), for all animals (*dots*, $n = 8$). Box and whiskers denote 25–75, and 2.5–97.5 percentiles respectively, midlines are means, notches are 95% confidence levels. Gray lines as in (**a**), but averaged over animals. **c** Population responses projected on the DV subspace of context (magenta line) and choice decoders (yellow line) in individual trials (*dots*) and their estimated normal distribution (mean and 2 std, *shaded ovals*) in different task contexts (*left* and *right panels*) and with different choices the animal made (*orange* and *red* for 'lick' and 'no-lick', respectively) for an example animal. Since the choice decoder changes across task contexts, different DV subspaces are used (*left* and *right* panels for 'attend' visual and 'ignore' visual context, respectively). Distributions show the marginal of population responses (gray lines indicate the distributions for the opposite context). **d** Histograms of the angle between context and choice DVs across animals (*dots*) in the two task contexts (*left* and *right*).

dependent decision making in non-human primates, the prefrontal cortex (PFC) was shown to receive unfiltered visual information which undergoes differential dynamics based on the appropriate context[35]. The study argued against the alternate scenario where top-down input acts within V1 to select the contextually appropriate input and relays only those to PFC. While in our study context was uncued and was therefore required to be maintained across multiple trials, our results provide additional insight into this question: contextual information shapes the activity in V1, it does so in a subspace orthogonal to the representation of stimuli and leaves stimulus-related V1 dynamics intact. This is also consistent with previous work showing context-dependent changes in the stimulus selectivity of population responses in mouse V1[6,36] or context-dependent modulation of V1 firing during navigation[37].

The context-dependent changes we identified in V1 population responses can be interpreted as a source of noise correlations. Our results show that these changes are orthogonal to the stimulus dimension and therefore do not contribute to information limiting correlations[38]. A recent study in non-human primates has highlighted that a feedback-driven component of noise correlation that is measured within a given task context changes across tasks and displays a structure reminiscent of information limiting correlations[39]. Thus, noise correlation measured within a task context and across task contexts might indicate different feedback components and their relationship can reveal the exact mechanism of context-dependent modulations[40,41]. Such context-dependent changes in noise correlations are not constrained to V1 but can be identified in MT and IT as well[42–44]. Recently, analysis of the choice-

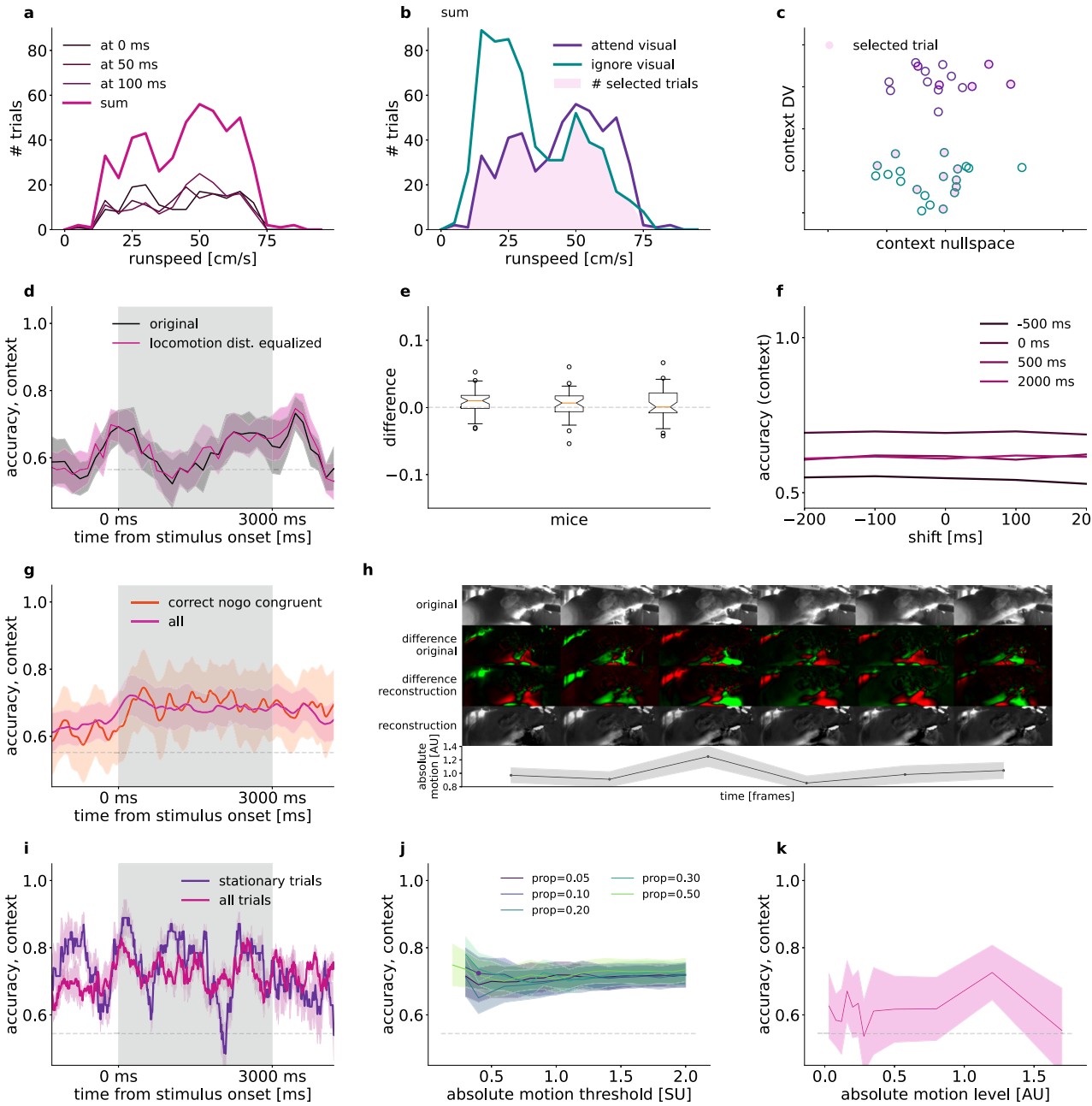

**Fig. 7 | Controlling for locomotion intensity differences between task contexts.**
**a** Example histograms of running speed at three time points during the trial (*thin lines*) and summary histogram over the time points (*thick line*). Histogram shows trials from the attend visual context. **b** Task-context specific running speed histograms. Histograms show summary histograms. Running speed distributions in the two contexts are matched by subsampling trials (*shading*). **c** Schematics illustrating run speed matching in the population activity space. Trial-by-trial population responses (*circles*) on a subspace spanned by the context DV and a direction perpendicular to that. After subsampling trials (*filled circles*) the distribution of running speed is equal in the two contexts (*filled circles*). **d** Mean context decoder accuracy of 10 locomotion matched subsamplings (*magenta*) and randomized control with same trial numbers (*black*). Shading shows 2 s.e.m. Gray dashed line indicates shuffled baseline. **e** Difference between run speed-matched and control decoder accuracies in a population of mice. Whisker plots indicate distribution over time ($n_T$ = 596 timepoints, whiskers between 5–95%, box between 25–75%, notch: median, 95% CI, flyers include all points). **f** Matched accuracies at single time points along

the trial (colored lines), with run speed shifted relative to neural activity (*horizontal axis*), example animal. **g** Accuracy of context decoding using all trials (*magenta*) and correct 'no-go' congruent trials only (*red*) at each timepoint, mean of n = 8 mice (line) and s.e.m. over mice + mean s.e.m.-s from cross-validation variance (bands). **h** Reconstruction from motion principal components. 6 subsequent frames (*left to right*): Original capture (*top*), smoothed difference from a moving average background (upper *middle*), reconstruction of difference from 40 PCs (*lower middle*), cumulative sum of the difference reconstruction (*bottom*). Absolute motion, i.e. mean of absolute values of 40 PCs, and s.e.m (*bottom graph*). **i** Accuracy of context decoding using all trials (*magenta*) and stationary only trials (*purple*) at each timepoint, s.e.m. over 10-fold cross validation (*bands*). **j** Mean context decoder accuracy over the trial time course of stationary trials at different absolute movement thresholds (*horizontal axis*) and proportion of timepoints allowed to exceed the threshold (*colored lines*). Mean s.e.m. over the time course (*bands*). **k** Mean accuracy over the time course from context-equalized number of trials at various absolute motion level intervals (*line*), and s.e.m (*band*).

related activity in MT of non-human primates demonstrated that much of the choice-related activity is orthogonal to the stimulus-induced variance[45,46]. Our results are consistent with this view and highlights that task-relevant variables partition the activity space into orthogonal subspaces.

We found that the context signal undergoes a transformation during the transition between off-stimulus and on-stimulus conditions but remains relatively stable within the conditions. This observation can be puzzling since the contextual variable is invariant across the conditions. Such transformation can be interpreted as changing fixed points that the network converges to in the two conditions. Recent work on recurrent dynamics in networks has started to uncover the properties of condition-dependent fixed point dynamics, and low dimensional dynamics in independent subspaces[47]. In our study, the transformation of the context representation could not be traced back to a simple dynamics in which a subpopulation of neurons switched their activities between conditions since a set of neurons displayed stable contribution to the context decoder throughout the trial. Our findings show that while it is possible to read out from the population activity the task being performed throughout the trial with simple linear decoders, only a select number of neurons is appropriate to construct a decoder that works equally well during and before task execution.

V1 activity is strongly modulated by locomotion[5,48,49]. In fact, recent studies suggest that the majority of stimulus-unrelated activity in V1 is associated with locomotion and movement[12]. Importantly, this study also implied that this stimulus-unrelated component of activity shares the subspace with off-stimulus activity. This subspace shared a single linear dimension with stimulus-evoked activity. We demonstrate that beyond movement-related signals, signatures of task context can also be identified in the off-stimulus activity. We found a similar orthogonal relationship between the subspace occupied by task context and that of visual stimuli (albeit our stimuli were simpler). In order to exclude the possibility that the task context signal identified in the analysis is contaminated by variations in locomotion patterns we performed a range of control analyses and confirmed that the identified task context representation was unaffected by variations in variations in a range of instructed and uninstructed movement patterns. We also demonstrated using video capture that context is invariantly represented in V1 in a number of control conditions: 1, when solely selecting stationary trials; 2, when lick' trials are excluded; 3, when trials are subsampled to equalize the distributions of movement intensities in various body parts. On the time scale of the context variable, other factors, such as success rate related changes in stress or frustration, might undergo modulations. However, trial-by-trial modulations in the context signal, as revealed by stronger context signal before an upcoming successful choice, modulation of the context signal within the trial, and relatively balanced performance in the two contexts for many of the recorded animals makes these factors less likely to account for the range of observations related to the context signal we identified.

How does contextual information reach V1? As V1 receives top-down inputs from anterior cingulate[50,51], motor, premotor, retrosplenial[52], posterior parietal[53,54], and higher visual cortices[55], as well as ascending inputs from lateral posterior thalamic nucleus[56], multiple direct and indirect inputs could contribute to contextual information in V1. An indirect route from the medial prefrontal cortex through the basal ganglia[57] can inhibit irrelevant information[58]. Besides these pathways, auditory information can reach V1 directly as well[9,59–61]. Ultimate answer to how contextual information is built up in V1 will require recordings from and causal manipulation of neurons projecting to V1 during the task. Chronic recordings from these areas will be critical for understanding how these representations emerge across days. Novel computational tools to track uninstructed changes in behavioral strategies[62] are critical for a deep understanding of the computations taking place in cognitively demanding tasks such as the set shifting task investigated here.

## Methods

### Surgery
All experimental procedures were approved by the University of California, Los Angeles Office for Animal Research Oversight and by the Chancellor's Animal Research Committees. 7–10 weeks old male and female C57Bl6/J mice were anesthetized with isoflurane (3–5% induction, 1.5% maintenance) ten minutes after intraperitoneal injection of a systemic analgesic (carprofen, 5 mg/kg of body weight) and placed in a stereotaxic frame. Mice were kept at 37 °C at all times using a feedback-controlled heating pad (Harvard Apparatus). Pressure points and incision sites were injected with lidocaine (2%), and eyes were protected from desiccation using artificial tear ointment. The surgical site was sterilized with iodine and ethanol. The scalp was incised and removed, and a custom-made lightweight omega-shaped stainless steel head holder was implanted on the skull using Vetbond (3 M) and dental cement (Ortho-Jet, Lang), and a recording chamber was built using dental cement. Mice recovered from surgery and were administered carprofen for 2 days, and were administered amoxicillin (0.25 mg/ml in drinking water) for 7 days. Mice were then water-deprived and trained to perform the behavior (discussed below).

Approximately 24 h before the recording, mice were anesthetized with isoflurane, a small craniotomy (0.5 mm diameter) was made above the right cerebellum and a silver chloride ground wire was implanted within the craniotomy and fixed in place with dental cement. A circular craniotomy (diameter = 1 mm) was performed above the right V1 (V1 targets were determined by regression of adult brain lambda-bregma distances: 1.7–2.5 mm lateral and 0.0–0.5 mm rostral to lambda). The exposed skull and brain were covered and sealed with a silicone elastomer sealant (Kwik-Sil, WPI). On the day of the recording, the mouse was placed on the spherical treadmill and head-bar fixed to a post. The elastomer sealant was removed and the craniotomy chamber was filled with cortex buffer containing 135 mM NaCl, 5 mM KCl, 5 mM HEPES, 1.8 mM CaCl$_2$ and 1 mM MgCl$_2$.

### Behavioral training
Following implantation of the headbars, animals recovered over 3 days, and received 10 to 20 min of handling per day, thus habituating the animals to human interaction for 4 days. Animals were then water-deprived, receiving approximately 1 mL of water per day. During this time, animals were placed on an 8-inch spherical treadmill (Graham Sweet) in the behavioral rig for at least 3 days to habituate to head-fixation for 15 min per day. The spherical treadmill was a Styrofoam ball floating on a small cushion of air allowing for full 2D movement (Graham Sweet, England). The animal's weight was measured daily to ensure no more than approximately 10% weight loss.

Animals were first trained to perform unimodal visual and auditory 'lick'/'no-lick' ('go'/'no-go') discrimination tasks. Licks are detected by using a lickometer (Coulbourn Instruments). Lick detection, reward delivery and removal, sensory stimulation and logging of stimuli and responses were all coordinated using a custom-built behavioral apparatus driven by National Instruments data acquisition devices (NI MX-6431) controlled by custom-written Matlab code. A 40-cm (diagonal screen size) LCD monitor was placed in the visual field of the mouse at a distance of 30 cm, contralateral to the craniotomy. Visual stimuli were generated and controlled using the Psychophysics Toolbox[63] in Matlab. In the visual discrimination task, drifting sine wave gratings (spatial frequency: 0.04 cycles per degree; drift speed: 2 Hz; contrast: 100%) at 45°, moving upwards, were paired with a water reward. Drifting gratings of the same spatial frequency but at 135° orientation, moving upwards, signaled a reward would not be present, and the animal was trained to withhold licking in response to the

stimulus. The inter-trial interval was 3 s, except for trials in which the animal had a miss or false alarm, then the inter-trial interval was increased to 6.5 s. The animal's behavioral performance was scored as a $d'$ measure, defined as the z-score of the hit rate minus the z-score of the false alarm rate, where z-score is the inverse cumulative function of the normal distribution, converting a probability to units of standard deviation of the standard normal distribution, with smallest allowed margins of 0.01 and 0.99 rates. Once animals reached expert performance ($d' > 1.7$, $p < 0.001$ as compared to chance performance, Monte-Carlo simulation), they were advanced to learning the auditory discrimination task where a low pure tone (5 kHz, 90 dB) indicated that the animal should lick for reward and a high tone (10 kHz, 90 dB) indicated that the animal should withhold licking. The inter-trial interval was similarly 3 seconds and the inter-trial interval was increased to 9 s after misses or false alarms. After animals learned the auditory discrimination task ($d' > 1.7$) they were trained to perform the multimodal attention task. In this phase, animals first performed one block of visual discrimination (30 trials). If their performance was adequate ($d' > 2.0$, correct rejection rate > 70%, hit rate > 95%) they then performed the visual discrimination task with auditory distractors present (the high or low tones) for 120 trials. Then, after a five-minute break, they performed the auditory discrimination task for 30 trials and if their performance was adequate ($d' > 2.0$, correct rejection rate > 70%, hit rate > 95%), they performed auditory discrimination with visual distractors present (oriented drifting gratings at 45° or 135° described previously). During each training day and during the electrophysiological recordings, each trial set started with 30 trials where only visual or auditory stimuli were delivered which signaled whether the animal should base its decisions on the later multimodal trials to visual or auditory stimuli respectively. Each trial lasted 3 s. When the cue stimulus instructed the animal to lick, water (2 μl) was dispensed two seconds after stimulus onset. No water was dispensed in the no-lick condition. To determine whether the animal responded by licking or not licking, licking was only assessed in the final second of the trial (the response period). If the animal missed a reward, the reward was removed by vacuum at the end of the trial. Animals performed 300–450 trials daily. Only one training session was conducted per day with the aim to give the animal all their daily water allotment during training. If animals did not receive their full allotment of water for the day during training, animals were given supplemental water one hour following training. Whether the animal started with the attend- visual or ignore-visual trial set was randomized. Importantly, the monitor was placed in exactly the same way during the auditory discrimination task as it was placed during the visual discrimination task, and a gray screen, which was identical to that during the inter-trial interval of the visual discrimination task and isoluminant to the drifting visual cues, was displayed throughout auditory discrimination trials. As a result, the luminance conditions were identical during visual and auditory discrimination trials.

### Motion detection

Head-fixed animals run on a treadmill consisting of an 8-inch Styrofoam ball (Graham Sweet) suspended 2 mm above an 8.5-inch Styrofoam cup (Graham Sweet) using pressurized air. Mouse treadmill rotation was recorded as an analog signal, using a custom printed circuit board based on a high sensitivity gaming mouse sensor (Avago ADNS-9500) connected to a microcontroller (Atmel Atmega328). The signal was initially recorded along with electrophysiology at 25 kHz, then down-sampled to 1 kHz, and low-pass filtered <1 Hz with a first order Butterworth filter. The processed signal was treated as a proxy of velocity. 6 animals were available for running speed measurements.

Infrared light and filter were placed in the box facing the mice. A camera (BFLY-U3-23S6M-C, FLIR) with zoom lenses (Zoom 7000 18–108 mm, Navitar) acquired images synchronized with TTL pulses triggered by FG085 miniDDS square function generator, at 20 frames/

s. The stream was saved as 8 bit grayscale video of 1980 × 1024 pixels. The side of the animal, where both large body, ear, and paw movements, as well as whisker and licking movements were visible in a cropped frame of 1398 × 574 pixels.

### In-vivo electrophysiology recordings

Extracellular multielectrode arrays were manufactured using the same process described previously[64]. Each probe had 2 shanks with 64 electrode contacts (area of each contact 0.02 μm²) on each shank. Each shank was 1.05 mm long and 86 μm at its widest point and tapered to a tip. Contacts were distributed in a hexagonal array geometry with 25 μm vertical spacing and 16–20 μm horizontal spacing, spanning all layers of V1. Each shank was separated from the other 400 μm. The electrodes were connected to a headstage (Intan Technologies, RHD2000 128-channel Amplifier Board with two RHD2164 amplifier chips) and the headstage was connected to an Intan RHD2000 Evaluation Board, which sampled each signal at a rate of 25 kHz per channel. Signals were then digitally band-pass-filtered offline (100–3000 Hz) and a background signal subtraction was performed[64]. To ensure synchrony between physiological signals and behavioral epochs, signals relevant to the behavioral task (licking, water delivery, visual/auditory cue characteristics and timing, and locomotion) were recorded in tandem with electrophysiological signals by the same Intan RHD2000 Evaluation Board.

After performing the visual and auditory cross modal tasks, drifting gratings were presented to map the orientation selectivity of each recorded cell. A series of drifting gratings of 6 orientations spaced by 30°, with both directions, randomly permuted, temporal frequency = 2 Hz, spatial frequency = 0.04 cycle per degree, contrast = 100%, was presented for 3 seconds, with a 3 second inter-trial interval, with each different presented 10 times.

### Acute microprobe implantation

On the day of the recording, the animal was first handled and then the headbar was attached to head-fix the animal on the spherical treadmill. The Kwik-Sil was removed and cortex buffer was immediately placed on top of the craniotomy in order to keep the exposed brain moist. The mouse skull was then stereotaxically aligned and the silicon microprobe coated with a fluorescent dye (DiI, Invitrogen), was stereotaxically lowered using a micromanipulator into the V1 (relative to lambda: 1.7–2.5 mm lateral and 0.0–0.5 mm rostral). This process was monitored using a surgical microscope (Zeiss STEMI 2000). Once inserted, the probe was allowed to settle among the brain tissue for 1 h. Recordings of multiple single-unit firing activity were performed during task engagement (approximately 1 h). After the recording, the animal was anaesthetized, sacrificed, and its brain was extracted for probe confirmation.

### Data analysis

**Behavioral measures.** To quantitatively characterize the performance of the animals, a four component sliding window average measure was calculated through the session. The width of the window was carefully optimized to 20 trials, to be long enough to capture consistent behavior in consecutive trials (blocks), while short enough to be sensitive to changes in behavior. The components were constructed for 'go' and 'no-go' signals, each having two for congruent (distractor would indicate the same action as the active signal) and incongruent (distractor signal would indicate opposite of the active signal) trials. The measures were calculated separately in the two contexts, thus including four possible combinations for both audio and visual context. The width of the sliding window was optimized to always include all four types of trial combinations, while sensitive to response strategy switches of the animal; we found that a 20-trial window adequately satisfied these criteria. Consistent and exploratory trials were defined as when all four moving averages were above or below chance (0.5), respectively. Thus,

e.g. a trial would be exploratory, if any of the congruent 'go', incongruent 'go', congruent 'no-go' or incongruent 'no-go' moving averages dropped to 0.5 or below. We modeled choices of mice during consistent trial blocks via fitting various models to the $p$ parameter of Bernoulli distributions. First, to rule out that animals based their choices on one of the modalities throughout the entire session, we constructed a simple model with opposite modality targets for $p$ (i.e. $p = 1$ for audio signals in visual context and vice versa), comparing it to correct targets (i.e. $p = 1$ for visual signals in visual context and vice versa). Then we analyzed potential context unaware and non-context related biased or random choices of animals. To assess a context-unaware strategy, we fitted a single parameter $p$ on the responses of the animal, which characterized the tendency of the animal to lick at the end of the stimulus presentation. We formulated two context-aware strategies, both characterized with a single parameter. In the bias model the 'go' trials were characterized by a certain behavioral response, while 'no-go' trials were fitted with bias, β, that characterizes the tendency of lick in trials when the animal was expected to withhold lick. In the lapse model both 'go' and 'no-go' trials were characterized by an error rate of λ, therefore the probability of lick in 'go' trial was 1-λ, while in 'no-go' trials it was λ. The $p$ values of all models were clipped between 0.001 and 0.999. Models were compared by their mean log likelihoods over consistent incongruent trials. We also generated 100 000 random single-context sessions of 70 trials with responses from the mean lick rate model at 75% lick rate, typical for incongruent trials in the weaker performing context of animals. We then applied the same moving average consistency criteria we used for mice. Then we counted the number of consistent trials ($N$), and the length ($L$) of all consecutive consistent blocks occurring at least once. We then calculated the approximate probability of and the cumulative probability of having at least $N$ consistent trials and $L$ consecutive length. For reference, when applied in two contexts, the calculated probabilities need to be raised to the square.

**Single unit activities (SUA).** Spike sorting was performed by Kilosort2[17], and then manually curated in phy2 using MATLAB and PYTHON to yield single unit activities, with consistency criteria for autocorrelograms, interspike interval histograms, waveforms, maximal amplitude electrode locations, low false positive or missed spikes and also stable feature projections throughout the recording session. Highly similar clusters were merged manually if cross-correlation revealed identical refractory periods and if interspike interval histograms and feature distributions matched to provide a resulting unit without drift signs. Clusters were split when PCA feature space and interspike interval histogram showed mixtures of stationary distributions, and the cross-correlograms improved. Putative excitatory and inhibitory cell types were distinguished based on spike waveform characteristics corresponding to broad and narrow spiking. We applied a gaussian mixture model to cluster in the two dimensions of time from trough to peak and amplitude ratios of trough and peak on all cells from all mice.

**Exclusion criteria to control for drift.** We took great care that only units showing no statistically identifiable drift in firing responses were included in the analysis. For this, signal-to-noise ratios of spike events were tracked during the parts of the recording session when the animal performed the task. For any given unit, drift was identified using a set of criteria on a long timescale. Specifically, we assessed the quality of the unit based on (A) unit separability in feature space, (B) stationarity of signal to noise ratio (S/N) and (C) stationarity of firing rate. We excluded units that did not meet our criteria.

**Spike counts.** Spike counts were calculated in sliding windows in 10-ms bins and smoothed using a symmetric Gaussian kernel ($\sigma = 100$ ms, optimized for linear decoding and typical firing rates). The kernel method approximates single trial instantaneous rate (IR). IRs were

used in two distinct ways. First, we used IRs to assess absolute instantaneous firing rates (FR) in Hz. Second, for methods requiring standardized input, IRs were transformed to z-scores, with mean calculated from prestimulus (from −1500 ms to 0 ms) time-averaged baseline activities, while standard deviation was calculated for the whole trial. Sensitivity of single units to task variables was assessed by the largest mean rate difference between trial averages during different task epochs.

**Neural activity vector.** Neural activity vector (referred to as population activity or activity) is defined as baseline-standardized SUA IRs as components of a time-varying vector. Neural activity vector space, specifically using the above coordinate system as basis, is defined as the space of possible activity patterns the neuron population can take. A single point in this neural activity vector space corresponds to IR for each neuron, while trajectories describe dynamics of activity.

**Individual cell sensitivity to task variables.** Sensitivity was defined as the difference between mean firing rates at the two variable values (left and right gratings for visual, low and high tone for audio, attend visual and audio for context, 'lick and 'no-lick for choice) divided by the standard deviation of the firing rates of the cell over all trials, averaged over all timepoints separately during pre- and on stimulus.

**Relative signal variance.** Total variance for signals with multiple conditions can be decomposed into the sum of noise variance and signal variance. Noise variance is the expected value over the conditions of the variances of the signal at specific conditions. Signal variance is the variance over the conditions of the expected values of the signals at specific conditions.

$$\mathrm{Var}_y[y] = E_x[\mathrm{Var}_{y|x}[y|x]] + \mathrm{Var}_x[E_y[y|x]]^2,$$

where x runs over two values of the condition, e.g. 45° and 135° visual stimulus, and y runs over the trials with y | x signifying trials conditioned on the indexed values of x, e.g. only 45° trials. Relative signal variance is a ratio of the signal variance and total variance, i.e. the second term on the right hand side divided by the term on the left hand side of the equation.

## Principal component analysis

We concatenated neural activity of all time points along a trial for all trials, capturing both trial to trial and within trial variance. Principal component analysis was performed to obtain orthogonal directions by ordered variances of population activity. Neural activity was projected to various subspaces defined by a number of principal components (PC) by dot products with PC unit vectors. Relative variances were shown cumulatively for activities projected on a growing subset of PCs. Thus we defined relative signal variance to extend the fraction of variance explained by the signal to a series of fraction of explained variances at multiple projections onto growing PC subspaces. To establish baseline signal variances of task variables, we repeated this analysis with the mean variance of 20 random orthogonal projections of the same dimensions as the increasing number of PCs.

## Linear decoding

We regressed neural activity with task variables using predictive trial-based cross-validation. Input to the decoder was a matrix with observations in different rows and features in columns. Observations were trials, each labeled with task variables such as stimulus identity, task context, and choice of the animal (licking or witholding licking). These identified nominal variables each consistently grouped trials into their respective two classes. Specific decoders were trained by conditioning on some of the non-decoded variables: for instance, on Fig. 5 we only selected successful trials for training the decoder. Feature space

representing activity of all units in a time-segment was constructed by concatenating for each unit 5 consecutive 10 ms width spike count data points spanning 50 ms, and concatenating these vectors for each unit. The time resolution of the IFR kernel slide and the time-width of the decoder were optimized to saturate information transfer from raw data while preserving high time resolution. Classification of trials was trained with logistic regression. Separate decoders were trained for different time points of a trial with 10 ms resolution. Decoders used all single unit activities available, with the expectation that discrimination-irrelevant features would be averaged out with a Gaussian noise model in the log odds space. Unless otherwise noted, all decoders were performed with class-stratified 10-fold cross-validation. Decoder accuracy figures show means and two standard errors of means of the 9 CV test runs for each timepoint. Averages across mice only use the means, boxplots show animal population mean and 2 s.e.m. for notches and 25–75, 5–95 percentiles for the boxes and whiskers.

Cross-tested decoders were introduced to assess how a decoder trained in one condition performs in another condition. A decoder is trained at one particular time point of the trial and its performance is tested at a different time point. To limit computational overhead, for these time point cross-tested decoders were trained with a cross validation scheme of 2 times 3:1 randomized train test splits instead of the default 10-fold cross-validation. These were accurate enough, but consumed significantly less computation resources than full 10-fold CVs. Accuracy decay rate was calculated at each training point by fitting a linear function on the decoder performances in consecutive time points of cross-tested decoders and taking the slope over the first 500 ms of forward test shifts. Angles between time course-shifted (t1–t2) decoders were calculated as $\gamma_{t1,t2} = \arccos \mathbf{d}^{(t1)}\mathbf{d}^{(t2)}$, $\mathbf{d}$ normalized at all t-s. The dimension of $\mathbf{d}$-s when finding orthogonal vectors are irrelevant as we assume the unimportant directions are random, thus highly likely to contribute zeros to the scalar product, as well as their noises cancel out to 0. Thus interesting angles are indistinguishable in N and 2 dimensional spaces. Average angles were calculated for each mice by averaging over the angle values in the lower triangles of prestimulus (−1500 to 0 ms) and on stimulus (0–3000 ms) block matrices, and the rectangular cross matrix (t1 = −1500 to 0 ms, t2 = 0 to 3000 ms). To reduce overlapping effects within the instantaneous firing rate kernel width (100 ms) and the feature space width of 50 ms, angle-matrix elements closer to 100 ms from the diagonal (|t1−t2| <100 ms) were discarded. We calculated average accuracies from the same matrix elements of the across-time decoder accuracy matrix as the angle matrix, excluding 100 ms wide band near the diagonal for pre and on.

We calculated effective chance levels for the data by averaging over 40 independent decoder cross-validation accuracy distributions as described above, with fully randomized trial labels for the multimodal blocks taken from a Bernoulli ($p = 0.5$) distribution. Thresholds indicate trial time course-averaged mean + 1 s.e.m of 40 runs of the means + 1 s.e.m.-s calculated over CVs. Decoders are expected to differ from chance only towards higher accuracy, thus the confidence level is one-sided 84% for 1 s.e.m. These were shown throughout multiple figures and panels as gray levels.

The consistent and exploratory blocks-restricted context decoders were constructed as follows: We included animals where both contexts had at least 10 consistent trials. The number of trials available to the decoders were equalized to the lowest number available in the four trial type combinations of consistent and exploratory trials and in any of the two contexts. We varied the number of trials via 20- repetition subsampling for the other three combinations for each animal, so that both classes have the same number of trials, even for the performance category with the larger trial pool. We measured decoder accuracies of the averages of these bootstraps. For these binary classification decoders, we employed leave-one-out cross-validation,

therefore the cross-validation variance is larger than for decoders with fewer cross-validation attempts. We compared the accuracy of the consistent and exploratory subset by subtraction. We excluded timepoints, where the accuracies of both decoders were below chance. We recalculated the effective chance level for the used number of trials for this criteria. Each mouse yielded a distribution of these accuracy differences. We concatenated the accuracy differences of all mice and all timepoints to establish a robust population level accuracy difference distribution.

### Assessment of the contributions of neurons to the decoder
The number of units recorded and successfully identified, and units that do contribute to the performance of a decoder varied across animals. Accuracy of decoders generally increases with the number of available neurons. We used partial correlation when comparing accuracies of separate decoders across animals to control for the number of units available. We performed linear least squares fit for the number of neurons predicting accuracy of decoders (Supplementary Fig. 2a, b). The residuals of these fits contain no information about the correlation between number of neurons and the accuracy of decoders. When such residuals of decoders that were fit to different conditions in the same animal correlated with each other (Supplementary Fig. 2c), the resulting coefficient is highly unlikely to be confounded by the number of neurons.

### Multidecoder subspace analysis (MDSA)
A linear two-way classification decoder, e.g. logistic regression, defines the Decision Vector (DV) as the optimal one-dimensional subspace of the neural activity vector space along which neural activity realizations in single trials are best separated for the classes in question, e.g. 45° and 135° visual stimuli trials. In other words the DV is the vector perpendicular to the N-1 dimensional hyperplane that best separates the two classes (also called the normal vector of the decision boundary). We calculated DVs by averaging over the 10 cross-validation runs, and averaging the DVs of a number of linear decoders trained at different time points of the trial: Unless otherwise noted, DVs are time averages of the first 1500 ms of the trial after stimulus onset.

Since the coordinates of all decision vectors are in the same neural activity vector space, their angular difference, $\gamma_{1,2} = \arccos \mathbf{d}^{(1)}\mathbf{d}^{(2)}$, where $\mathbf{d}^{(k)}$ is the normalized DV for the k-th decoder, provide a meaningful description of neural activity regarding task relevant variables the decoders were trained to discriminate.

Since decision vectors, $\mathbf{d}^{(k)}$ are defined in the same space (the population activity space), individual decision vectors can be combined together as a new basis spanning a multidimensional task-relevant subspace, $S$:

$$S = \mathrm{span}(\mathbf{d}^{(1)},\mathbf{d}^{(2)},\mathbf{d}^{(3)}, \ldots).$$

This subspace will contain task-relevant neural activity the linear decoders are able to pick up, and generally will be low-dimensional in our experimental paradigm. These bases do not necessarily form an orthogonal coordinate system, though: the basis vectors are linearly independent, but not necessarily orthogonal. Such skewed coordinate systems appear when some task variables have dependencies between each other, e.g. correct animal choice and stimulus identity of the relevant modality should be heavily dependent on each other if the animal performs the task well. We normalized the decision vectors to show each coordinate with unit length bases (SU, standardized units).

A low-dimensional representation of the high-dimensional neural population activity can be obtained by projecting the population activity onto the task-relevant subspace. The relevant bases can include both a set of principal components (PCs) of a PCA, or

subspaces of single or multiple DVs as basis:

$$\mathbf{u} = \mathbf{P}\,\mathbf{a},$$

where **a** is the N dimensional activity vector, **u** is K dimensional subspace-specific activity, where K is defined by the number of PCs or DVs used as basis, and **P** is a K by N projection operator. **P** was defined by the K rows of transposed PC or DV matrices, respectively. Projection operators were normalized. We projected either each time point individually onto the projection bases, yielding full trajectories: $\mathbf{u}(t) = \mathbf{P}\,\mathbf{a}(t)$, or projected time-averages of an interval within a trial: $\mathbf{u} = \mathbf{P} <\mathbf{a}(t)>_t$. Then the resulting task-relevant activity projection, $\mathbf{u}(t)$ or $\mathbf{u}$, was conveniently visualized in an orthonormalized version of the coordinate system of the subspace $S$ by the Gram-Schmidt procedure over **P** showing the first few basis in **P** as orthogonal as possible in orthonormal coordinates. Bases were reordered for each question we asked to project the most relevant variables examined onto the most natural first few orthogonal coordinates. The PCA basis was orthogonal by construction.

When decoding from neural activity projected onto a subspace, the projection operator in the coordinates of the original neural activity basis is: $\mathbf{P} = \mathbf{D}\,(\mathbf{D}^\mathsf{T}\mathbf{D})^{-1}\,\mathbf{D}^\mathsf{T}$, where $\mathbf{D} = [\mathbf{d}^{(1)}, \mathbf{d}^{(2)}, \mathbf{d}^{(3)}, \ldots]$, a basis for $S$. In case of the 1 dimensional subspace of e.g. the visual decision vector, $\mathbf{D} = \mathbf{d}$, the projection $\mathbf{D}\,(\mathbf{D}^\mathsf{T}\mathbf{D})^{-1}\,\mathbf{D}^\mathsf{T}$ will become $\mathbf{d}\,\mathbf{d}^\mathsf{T} / \|\mathbf{d}\|^2$.

We projected activity to the nullspace of a DV with $\mathbf{I} - \mathbf{P}$, a reduced rank projection operator, where **I** is the identity and **P** is the above defined 1 dimensional projection operator, $\mathbf{d}\,\mathbf{d}^\mathsf{T}/\|\mathbf{d}\|^2$. With QR decomposition, $\mathbf{Q}\,\mathbf{R} = \mathbf{I} - \mathbf{P}$ we transformed $\mathbf{I} - \mathbf{P}$ into a coordinate system, **Q**, where the excess linearly dependent row with now all 0 values could be discarded for the projection operator. Calling also **R** with the row discarded, is an N-1 by N matrix. Thus the remaining activity when projected onto **Q**, $\mathbf{u} = \mathbf{R}\,\mathbf{a}$, was N-1 dimensional.

An additional cross-validation step is necessary when decoding a variable from activity projected onto its own 1D subspace. We defined outer cross-validation as fitting a decoder in a training set, and using the DV found in this training set as the basis to project activity onto the test set. Then the inner cross-validation on the projected activity in the outer test set was used with its own folding as described above. The resulting inner cross-validation accuracies were averaged over the outer folds yielding a single accuracy for each timepoint. We used an outer two-fold cross-validation scheme for context in this study.

## Locomotion distribution matching

In order to separate the possible contribution of locomotion from the contribution of cognitive components to the context signal, we performed run speed distribution matching between contexts. We calculated the running speed of animals from the discrete time derivatives of the magnitude of the vectorial sum of forward and lateral displacement of the spherical treadmill. Running speed histograms were constructed between 0 and 100 cm/s at 5 cm/s bins. We averaged the run speed in 50-ms time windows, matching the length of the time window used by the linear decoders. The run speed data along with the population activity data defined a joint distribution in which every single trial in a context contributed a single data point. To control for differences between contexts, the histogram of run speeds in a given context served as an approximation of the distribution of run speeds. This histogram, however, is a noisy approximation since the set of approximately 60 data points that corresponds to the individual trials in a given context suffers from sampling noise. To rectify this issue we expanded the set by including data points from consecutive time windows. As a consequence, we could provide a more reliable estimate of speed distribution albeit in the end at a lower time resolution of 150 ms. To have a matching running profile in the two contexts, we subsampled the running speed distributions such that the histogram was identical in the two contexts. Then using the trials

retained after subsampling, we performed the decoding for the context variable. We found that the increase in the number of trials using three consecutive 50 ms intervals balances out the number of trials we lose in all run speed bins, either from one or the other condition so that the decoders work on similar numbers of trials as without this matching on previous analyses. We averaged 10 of such matching subsamplings, and compared the cross-validated accuracy with the average accuracy of 10 randomized sub-samplings of the same number of trials for each run speed bin. Decoder results were such subsampling averages of means and 2 s.e.m.-s of the cross-validation accuracies. Shifts were defined as the difference between time indices within trials between runspeed and neural data, unsyncing the two, and the new full overlapping windows were used for analysis.

## Video motion analysis

In addition to running speed, we identified diverse movement patterns from video capture. Apart from using the raw pixel intensities as posture proxies, we also calculated a frame pixel intensity difference, $d$, from a moving average background with the equations: $b_t = \beta^* x_t - (1-\beta)^* b_{t-1}$ where x is a pixel from a frame, and $b$ is the background, and $d_t = \lambda^*(x_t - b_t) + (1-\lambda)^* d_{t-1}$. The rates appeared robust, we chose $\beta = 0.3$, $\lambda = 0.9$ for optimal combination of noise filtering, but still detecting high frequency movements.

Then we performed principal component analysis (PCA) using minimum rank singular value decomposition (SVD), $\mathbf{X} = \mathbf{U}\,\mathbf{S}\,\mathbf{V}^\mathsf{T}$, to get a small dimensional representation of the movement of the animal, from either the posture (*raw*) or the movement (*differential*) frame stream. Due to computer memory constraints, we first calculated the first 200 principal components based on the variance of 3000 frames chunks (chunk PCs, outer PCA). Then we concatenated all frames of each chunk PCs, and performed an inner PCA that yielded the first 40 principal components (PCs) over the variance of the chunk PCs from the entire session. We used these PCs to either reconstruct frames in the pixel space to cross check that the PCs actually capture movements, and to perform further analysis on the small dimensional movement space. For reconstruction we applied the inverse of the inner, then outer PCAs, multiplying the PC matrix with the $(\mathbf{V}^\mathsf{T})^{-1} = \mathbf{V}$ transformation matrix, while at both stages applying the inverse of the standardization and mean subtraction functions. We only reconstructed the original image from the differential by cumulatively adding together subsequent differential frame reconstructions. We scaled the resulting differential and cumulative reconstructions to be visually comparable with the differential frames and the original movie frames, respectively. Video reconstruction output was downsampled at the file input-output stage to one fifth.

An absolute posture or motion scalar for each frame was constructed by taking the absolute value of the principal components at each frame, taking their 4 backward 4 forward 9 frames moving average, and taking the mean over principal components. We further analyzed the differential stream. Stationary frames were defined as frames with a motion value lower than a threshold. Trials were defined as stationary if in not more than a set percentage of time points the absolute motion values exceeded stationary thresholds. Selected stationary trials were used from both contexts for the context decoders, to assess the effect of stationary movement patterns. Similarly, we binned trial time course points in intervals of absolute motion levels, excluding timepoints where less than 5 trials were available for any of the contexts, then equalizing the number of trials in the two contexts by subsampling trials from the more numerous context, and then averaged over time course points at each motion level bin.

Regions of interest (RoI) were chosen by visually cross-referencing multiple movement patterns across the video capture. Nose area also contained the whisker pad. Licking was restricted to the mouth area. Right forepaw occasionally moved to the ear or mouth area. A motion scalar was calculated from averaging over the pixels of

the absolute values in the smoothed difference frames for each RoI. Note, that this method did not use the global PCA, but rather the raw intensities after the frame differential. Equalization for context decoding was the same, as for the PCA derived absolute motion scalars. Random trials were chosen instead from the same motion level for the equalized shuffled control.

We calculated the variance predictively explained by using a 5-fold cross validated ridge regularization regression to predict neural activity from motion PCs or motion PCs from neural activity. We used posture PCs, motion PCs, and body parts motion intensities. Predictors were either single components, or the entire vector. In addition, reduced rank multilinear regression at decreasing rank was also attempted to reduce the number of parameters to fit. We attempted 20 Hz resolution, 1 Hz resolution, and either at each timepoint during trials separately, or concatenating all timepoints for all trials as separate observations. Iterations were deemed not reaching a fit, if the cross validated predictive $R^2 = 1 -$ residual sum of squares / total sum of squares, was lower than 0.

### Reporting summary

Further information on research design is available in the Nature Portfolio Reporting Summary linked to this article.

## Data availability

Behavioral and spike sorted electrophysiological data, and movement data are available in the Zenodo database: https://zenodo.org/record/7900224. Source data are provided with this paper.

## Code availability

Event generator and behavior data acquisition: https://github.com/golshanilab/AttentionTask.git Analysis: https://github.com/CSNLWigner/mouse-v1-context https://github.com/CSNLWigner/mouse-v1-context-movement.

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

## Acknowledgements
The authors thank Rachna Goli, Allison Foreman, Brandon Sedaghat, and Tara Shooshani for their help with training the animals to perform the behavioral task, as well as Zsombor Szabó, and Andrea Albert for their help with behavioral analysis. The authors would also like to thank Anne Churchland and Dario Ringach for their comments on a previous version of the manuscript. P.G. and G.O. were supported by a grant from the Human Frontiers Science Program, P.G. was supported by grants 1R01MH105427, R01NS099137, 1P50HD103557, M.A.H. and G.O. were supported by a grant by the Hungarian Brain Research Program (2017-1.2.1-NKP-2017-00002).

## Author contributions
P.O.P., M.E, and P.G. designed and optimized the behavior, and designed all experiments. D.T., M.E. and K.S. trained animals and performed the recordings. D.T, M.V.M., K.S. and M.A.H. curated the data. M.A.H. and G.O. designed the analysis. M.A.H. performed the analysis mainly with input from G.O., but also P.O.P and P.G. M.A.H., P.O.P, P.G. and G.O. wrote the manuscript.

## Funding

## Competing interests
The authors declare no competing interests.
