## [Peer Review File · Nature Communications]

Continuous multiplexed population representations of task context in the mouse primary visual cortexEditorial Note: This manuscript has been previously reviewed at another journal that is not operating a transparent peer review scheme. This document only contains reviewer comments and rebuttal letters for versions considered at *Nature Communications*.

REVIEWER COMMENTS

Reviewer #2 (Remarks to the Author):

I previously reviewed the exact same version of the manuscript in another journal. (Perhaps an older version is submitted in error?). My concerns regarding the manuscript remain the same and are listed below.

The revision of the manuscript has strengthened the manuscript and most concerns have been addressed. However, one major concern remains - regarding differentiating between a multi-sensory and contextual switching task (R2 pt 4 & raised by other reviewers too). Unless this can be shown convincingly, this appear to be a multi-sensory task with a bias for audition, rather than a context switching task.

First of all, even after excluding many animals, there is a very high false positive rate across all trials (given the <50% performance in Fig 1E), which suggests the animals are generally licking unless there are no-go congruent trials.

More specifically, the concern is related to how stringent the threshold is for being included in the 'high performance' period. Based on the described methods, there needs to be at least 1 trail of each of the four types within a 20-trial window and the four trial types need to be over 50%. Let's consider the visual incongruent sessions. Now, in a 20-trial window on average there should be 5 trials of each type. If the animal was licking at chance, and there were 2 trails with licks and 3 without, that would give a fraction correct of 0.6. In this case, even though it was random, the 20-trial window would be defined as 'high performance'. In a window with 4 trials, getting 3 correct (very likely if at chance level) would give 0.75 fraction correct and also high performance. So, both the 20-trial window and the 0.5 performance threshold need to be evaluated very carefully.

"The width of the sliding window was optimized to always include all four types of trial combinations, while sensitive to response strategy switches of the animal; we found that a 20-trial window adequately satisfied these criteria." – there should be evidence provided for any choice made, especially given how crucial it is to the interpretation of the results.

Minor points:

- Fig 1D: Since attend visual always follows visual and attend audio follows audio, the order in which the conditions are shown in this panel is misleading.

- Fig 1F: Instead of showing only the smoothed version, it would be useful to see what happened per trial... perhaps a raster of lick/no-lick in each condition.

- 'high-performance / low-performance' would be an overstatement in any case (see major concern above) and misleading, if its about being over chance. There should be a subtler more appropriate term used to define the two epochs.

Reviewer #4 (Remarks to the Author):

Below I comment on authors' responses to reviewer #1 that I feel still have issues:

2. The authors addressed the reviewer's point in different ways from what was suggested. They added two analyses to rule out the possibility that any behavioral differences explain the difference in neural activity between the two contexts. First, they attempted to relate population activity to behavioral features extracted from video recordings. They basically failed to build a model to predict population activity by behavioral features, and based on this failure, made a claim that difference in population activity between the contexts cannot be accounted for by behavioral features. However, I consider that as the authors stated, this is too indirect evidence and that in experimental science any argument cannot be made based on such a negative result, which can be obtained with a variety of reasons. Second, the authors selected trials where mice were stationary according to their video analysis and reported that the context decoding was still as accurate as that without sub-selecting trials. Although this does not directly address the reviewer's point, this at least supports the authors' argument and would be more convincing if they could show that the sub-selected trials are not biased to one of the trial types (go vs nogo) and in performance level between visual and auditory blocks. Also, Figure 7G should be presented in a better way.

4. The authors did excellent jobs to characterize task performance by defining congruent and incongruent trials. However, I have a strong concern that mice did not learn the cross-modal switching task as the authors wished for. First, Figure 1E clearly demonstrates that mice perform the task at chance level in incongruent nogo trials especially during visual blocks. This is the most important trial type to assess how well mice can switch modalities (given that water deprived mice are strongly biased for licking for reward, performance in go trials doesn't tell much). The data looks like it is hard for mice to withhold licking in response to the auditory go stimulus during visual blocks (i.e. they were still attending to auditory stimuli during visual context). Second, although there are some time periods in which mice performed this trial type at above chance level (shown with red dotted line in Figure 1F), such increases in performance levels are accompanied by increases in miss rate in congruent go trials

(green solid line), meaning that mice just started ignoring the auditory go stimulus irrespective of the accompanying visual stimuli (i.e. they are not really attending to visual stimuli in the visual context during these periods). The most straightforward interpretation would be that mice did not understand the modality-switching task and any contextual modulation of behavioral models (Figure 1G) and the population activity (rest of the figures, even the difference in context decodability between different performance levels) can come from any behavioral differences related to task performance (e.g. behaviors related to stress or frustration for not being rewarded) which the author failed to describe in this paper, but not from cross-modal attentional shift. It is up to the editors to decide if the orthogonal representation of visual stimuli and any unknown (not necessarily cognitive) variable is still interesting, but their argument that the contextual signals represent “cognitive” variables should be substantially toned down and they should comment on “behavioral” variables as potential caveat.

Also, the way how they calculated moving average of performance level in Figure 1F is misleading. For example, the red dotted line starts increasing even though the mouse is still making 100% false alarm decisions. Finally, I don't get the authors' following statement in their response: “We divided the sessions into different types of epochs: 1. High performance epochs : Blocks of trials when animals performed well during incongruent trials during both visual and auditory attention sections. 2. Low performance epochs: Blocks of trials when the animal performed well during incongruent trials only during either visual or auditory attention trials, but not both.” Given that visual and auditory context blocks are separated in time, how can such blocks of trials be defined? This is not the same as how high-performance trials were defined in the manuscript. Possibly I'm misunderstanding something...

9. The authors' response does not directly address the reviewer's point. At least to control for behavioral confound related to licking, the authors should redo the analysis by using only correct nogo trials and test if they still see the abrupt change.

Response to reviewers

We thank the reviewers for their thoughtful comments. Based on the comments we have updated the manuscript. Our responses can be found to individual comments below. For easier tracking of the updates, in the manuscript we have highlighted changes with blue.

R2

I previously reviewed the exact same version of the manuscript in another journal. (Perhaps an older version is submitted in error?). My concerns regarding the manuscript remain the same and are listed below.

We apologize to the reviewer for the confusion caused by this submission. Because the submission was new to this journal, we did not highlight the updates we made in response to the reviewer's comments. We also believe that our 'response to reviewers' document that outlined the actual changes did not reach the reviewer. In this new version we have clearly highlighted all the changes we have made. Further, compared to the earlier submission we have more extensively revised the manuscript. As part of this effort, we have extensively rewritten the first part of the results. These changes addressed reviewer concerns regarding streamlining the introduction of the paradigm, animal training and behavioral testing. Furthermore, we now present a more complete set of behavioral models to quantify how contextual modulation of responses can be identified in the behavioral data.

The revision of the manuscript has strengthened the manuscript and most concerns have been addressed. However, one major concern remains - regarding differentiating between a multi-sensory and contextual switching task (R2 pt 4 & raised by other reviewers too). Unless this can be shown convincingly, this appear to be a multi-sensory task with a bias for audition, rather than a context switching task.

We have now included multiple behavioral models to explicitly test alternative hypotheses about the strategies the animals might employ instead of contextual switching. Using these quantitative models we demonstrated that context-aware behavioral models consistently outperformed alternative models. This is now presented in Figure 1J,I and further discussed below.

First of all, even after excluding many animals, there is a very high false positive rate across all trials (given the <50% performance in Fig 1E), which suggests the animals are generally licking unless there are no-go congruent trials.

More specifically, the concern is related to how stringent the threshold is for being included in the 'high performance' period. Based on the described methods, there needs

to be at least 1 trail of each of the four types within a 20-trial window and the four trial types need to be over 50%. Let's consider the visual incongruent sessions. Now, in a 20-trial window on average there should be 5 trials of each type. If the animal was licking at chance, and there were 2 trails with licks and 3 without, that would give a fraction correct of 0.6. In this case, even though it was random, the 20-trial window would be defined as 'high performance'. In a window with 4 trials, getting 3 correct (very likely if at chance level) would give 0.75 fraction correct and also high performance. So, both the 20-trial window and the 0.5 performance threshold need to be evaluated very carefully. "The width of the sliding window was optimized to always include all four types of trial combinations, while sensitive to response strategy switches of the animal; we found that a 20-trial window adequately satisfied these criteria." – there should be evidence provided for any choice made, especially given how crucial it is to the interpretation of the results.

We thank the reviewer for providing a thoughtful example. To address the issue of a potential context-unaware decision strategy, we devised multiple behavioral models that we directly contrasted with context-aware versions of the behavioral model (Fig. 1I, J and related text in the main text). The context-unaware model is a generalization of the model that the reviewer refers to, in which a lick bias accounts for hits and errors. Please note that a model with lick bias covers all incongruent trials (both 'go' and 'no go') and cannot be constrained to incongruent 'no-go' trials since such a constraint would implicitly assume that the animal differentiates 'go' and 'no-go' trials which can only be done based on context. As such, on Fig. 1F, in the worst performance visual-context incongruent trials a lick bias should account for the difference between a single error in the 'go' trials and eight errors in the 'no-go' trials, thus with low explaining power.

We considered context-aware models that feature a single parameter. Using a single-parameter model ensures that the number of parameters, and thus the complexity of the model, is identical to the lick-bias model. Importantly, the simple context-aware models with either a bias or a lapse parameter was powerful enough to consistently beat the context-unaware models in all animals, depending on whether a given animal is described better with a small lick bias or lapse (also cf. Fig. 1H, red dashed lines). These results indicate that unstructured errors or learning a single task cannot account for behavior. We acknowledge though that deterioration of performance in incongruent 'no-go' trials is important. Based on the above results, a plausible interpretation alternative to unstructured errors might be that shifts in behavioral strategy can occur within a single context. Indeed, the tendency of licking under uncertainty in a go/no-go task is consistent with a strategy to gather information about the availability of reward or alternatively making decisions based on the irrelevant modality. An explicit statistical model for such shifts would rely on Hidden Markov Models to switch between strategies (e.g. Ashwood et al 2022, Nat Neurosci). The limited behavioral data collected in a session prevented us from directly using such a model in our analyses and we used the moving averages as a proxy for slowly shifting strategies instead. In the manuscript we analyze the behavioral models on consistent trials (formerly high performance trials, see our reply to this reviewer's point below). Such an analysis

ensures that even though strategies might shift, periods when the animal is performing the right strategy are identified.

In response to the reviewer's suggestion of a quantitative analysis of the moving average we used in establishing consistent blocks, we constructed a statistical test. Based on the number of trials in a given context and on the lick rate of the animal in the incongruent trials we generated a large set of synthetic data. We used the moving average that we applied on experimental data to identify consistent trials. Based on the consistent blocks identified in the synthetic data we constructed a histogram and measured the probability of the applied criterion ($N_{\text{consistent}} \geq 10$) and also the probability of having at least the same number of consistent trials actually identified at the particular animal ($N = 20$, Fig. R1).

Figure R1. Likelihood of identifying a given number of consistent trials in 70 trials similar to the number of trials in a given context. Likelihood is established by synthesizing a large set of trials from a generative model which produces licks at a rate measured at the example animal (~75%). The threshold we used (number of consistent trials is at least ten) has a likelihood of 0.0046. The number of consistent trials found at the example animal has a likelihood < 0.0002 .

In summary, our analysis demonstrates on an animal-by-animal basis that the context-aware model outperforms a context-unaware behavioral model. Note, however, that while consistent with these results, two animals that featured the lowest lengths of consistent blocks were removed from further analysis and the results was rewritten focussing on the eight animals in which the advantage of the context-aware model can be clearly demonstrated. The update of the cohort did not affect the results of the paper except for Supplementary Figure 5, in which the original finding could not be confirmed in the smaller cohort. As a consequence this figure and

corresponding text were removed. The removal of this Supplementary figure does not affect the conclusions of the paper.

Minor points:

- Fig 1D: Since attend visual always follows visual and attend audio follows audio, the order in which the conditions are shown in this panel is misleading.

Our goal was to overlay the performance in the two tasks so that these can be directly contrasted. We clarified this point in the text to avoid confusion. Also note, that the order of the attend audio and visual blocks in a multimodal session was randomized.

- Fig 1F: Instead of showing only the smoothed version, it would be useful to see what happened per trial... perhaps a raster of lick/no-lick in each condition.

Thank you for the suggestion. We have added markers indicating the outcomes of individual trials to the figure. We agree that this visualization contributes substantially to better assessment of potential interpretations. In particular, both congruent and incongruent trials display high levels of regularity in both contexts.

- 'high-performance / low-performance' would be an overstatement in any case (see major concern above) and misleading, if its about being over chance. There should be a subtler more appropriate term used to define the two epochs.

We have updated the text using a terminology that is agnostic about 'quality' of the performance and used 'consistent' and 'exploratory' blocks instead of high- or low-performance and now refer to these as consistent and exploratory blocks.

R4

Below I comment on authors' responses to reviewer #1 that I feel still have issues:

2. The authors addressed the reviewer's point in different ways from what was suggested. They added two analyses to rule out the possibility that any behavioral differences explain the difference in neural activity between the two contexts. First, they attempted to relate population activity to behavioral features extracted from video recordings. They basically failed to build a model to predict population activity by behavioral features, and based on this failure, made a claim that difference in population activity between the contexts cannot be accounted for by behavioral features. However, I consider that as the authors stated, this is too indirect evidence and that in experimental science any argument cannot be made based on such a negative result, which can be obtained with a variety of reasons. Second, the authors selected trials where mice were

stationary according to their video analysis and reported that the context decoding was still as accurate as that without sub-selecting trials. Although this does not directly address the reviewer's point, this at least supports the authors' argument and would be more convincing if they could show that the sub-selected trials are not biased to one of the trial types (go vs nogo) and in performance level between visual and auditory blocks. Also, Figure 7G should be presented in a better way.

We thank the reviewer for suggesting additional controls for movement invariance of the context representation we identify in neuronal activity. We added two analyses.

First, we examined context representation in correct 'no-go' trials. The Reviewer suggested that a balanced set of 'go' and 'no-go' trials would be desirable for motion controlled analysis. However, 'go' trials are characterized by higher motion intensity levels compared to 'no-go' trials, due to licking in 'go' trials, a movement pattern inherent in the design of the paradigm (new Suppl. Fig. 7F). Consequently, stationary trials contain fewer licking trials and balancing 'go' and 'no-go' trials proved to be statistically not feasible. Instead, we propose to constrain the analysis to all 'no-go' trials as follows. Note, that the analysis of contextual modulation requires that the stimulus statistics over trials are identical in the two contexts. Consequently, we relied on congruent trials, otherwise contextual modulation could not be separated from the effects of different stimuli. This introduced a limit on the number of trials that could be used in the decoding analysis. Considering the limitations on the trial numbers we decided not to use further stratification of the data according to motion intensities and decided to test context decoding in correct congruent 'no-go' trials regardless of movement intensity. A benefit of this design was that we could perform the analysis in all mice. The analysis resulted in identical context accuracy decoding to that obtained on all trial types. The analysis indicates that even if no licking was present, there was a difference in neural firing patterns between the two contexts. We include this analysis as a new panel (Fig. 7G).

Second, we designed a stricter analysis of movements. Instead of performing decoding on trials below a threshold, we constructed decoders for fixed intervals of motion intensity ranging from nearly stationary periods to intense motion periods. Bin boundaries were established such that the number of trials in each motion intensity level interval were approximately equal (new Suppl. Fig. 7G). Even after equalizing over the number of trials in the two contexts, each bin showed similar statistically indistinguishable predictive context decoder accuracies, but of course from a lower number of trials. This analysis indicated that irrespective of the overall movement intensity, contextual differences can be identified in neuron population activity.

We extended the video analysis by tracking specific regions of interest that were defined based on the body parts of the video-recorded animal (new Suppl. Fig. 8A). We calculated the mean absolute motion from differential frames for the different ROIs. We found in each body part that the absolute motion level between the two contexts is statistically similar (new Suppl. Fig. 8B). To ensure that marginal differences between contexts are not contributing to the observed context signal, we investigated if fully equalizing the distributions of movement intensity between

the two contexts affects context decodability. We performed equalization using balanced subsampling for individual ROIs separately as joint equalization could not ensure sufficient number of trials for decoding. We decoded context from these equalized trials. First, we found that context decoding accuracies from equalized trials were statistically similar between various body part regions. Second, they were statistically similar to accuracies from a control decoder where the same number of trials were sampled from all movement intensity levels (Suppl. Fig. 8C).

At this point we would like to reiterate our argument, that if any V1 neural pattern that still corresponds to movement differences remains after these analyses, its explained variance is certainly very small, while variance between the two contexts is rather large at 65-85% accuracy.

We have improved the visualization of movement analysis. Original frames showing reconstructions based on PCAs (originally shown on Fig. 7G) have now been moved to 7H, and we have improved visualization by arranging them so that they illustrate movement from left to right. We have also added the reconstructed sum as the fourth line (note, that it is based on smoothed differential, unlike the original top row), and indicate non-smoothed absolute motion level as a subplot below. Most importantly, we have selected a snippet of front paw movement during running as it better visualizes fading (red) and appearing (green) illumination on the differential motion maps. We have also added example filters of the dimension reduction to Suppl. Fig. 7H to demonstrate interpretability.

Finally, we would like to directly address the reviewer's comment on potentially building a model that could predict neural activity in V1 based on behavioral data or one that could predict behavioral data based on neural activity. Using cross-validation, we have built a model for regressing neural activity from movement, and movement from neural activity. We used both vector predictors and scalar predictors, as well as multivariate reduced rank regression to optimize for the number of predictors and targets. We took the Reviewer's comment seriously by extending our original analysis that was based on motion energy signals in PCs with a posture based analysis using raw PC intensities. Furthermore, we analyzed video of individual body parts by identifying ROIs in the videos and used the motion energy of body parts as regressors for neural data. We show example analyses for individual neurons and PCs in Fig. R2 (below). The figure indicates that predictive fits cannot be obtained on our data with any of these approaches. The analysis presented here is qualitatively similar to what we obtained for all the neurons and all the PCs. Models with higher number of parameters were more susceptible to capture noise structures, hence the higher training R^2 and consequently increasingly failed test R^2 . Fewer parameter models show less difference between train and test conditions. Note, that if the total sum of residuals (pointwise predictive errors) is larger than the total variance in the true data, R^2 will be negative.

Figure R2. Regressions between movement and neural activity. R^2 values and predicted data points are either for the train (blue) or test (orange) of the cross-validation. **Top left:** Explained variances (vertical axis) from separate regressions throughout the trial (horizontal axis) from a single body part to a single neuron. **Top right:** Predicted data points of a PC intensity (vertical axis) regressed from a single neuron (horizontal axis). **Bottom left:** Explained variance of a single PC regressed from all neurons throughout the trial. **Bottom right:** Explained variances from multivariate reduced-rank regression of PCs from neurons. Train (*top bundle*), test (*bottom bundle*). Colors range from 1 parameter (*yellow*) to the maximum number of parameters (*turquoise*).

Based on these analyses we speculate that the population of neurons recorded in our experiment failed to sample V1 neurons that would have shown a high level of predictive interaction between their activity and movement behavior. Note that in Stringer et al. 2019., Suppl. 7C shows neuron population size dependence of explained variance, and the smallest population size (2000 neurons) indicates 20% explained variance. The extrapolated value for the population size we have in our experiment seems to be close to 0%.

4. The authors did excellent jobs to characterize task performance by defining congruent and incongruent trials. However, I have a strong concern that mice did not learn the cross-modal switching task as the authors wished for.

First, Figure 1E clearly demonstrates that mice perform the task at chance level in incongruent nogo trials especially during visual blocks. This is the most important trial

type to assess how well mice can switch modalities (given that water deprived mice are strongly biased for licking for reward, performance in go trials doesn't tell much). The data looks like it is hard for mice to withhold licking in response to the auditory go stimulus during visual blocks (i.e. they were still attending to auditory stimuli during visual context).

Second, although there are some time periods in which mice performed this trial type at above chance level (shown with red dotted line in Figure 1F), such increases in performance levels are accompanied by increases in miss rate in congruent go trials (green solid line), meaning that mice just started ignoring the auditory go stimulus irrespective of the accompanying visual stimuli (i.e. they are not really attending to visual stimuli in the visual context during these periods).

The most straightforward interpretation would be that mice did not understand the modality-switching task and any contextual modulation of behavioral models (Figure 1G) and the population activity (rest of the figures, even the difference in context decodability between different performance levels) can come from any behavioral differences related to task performance (e.g. behaviors related to stress or frustration for not being rewarded) which the author failed to describe in this paper, but not from cross-modal attentional shift.

It is up to the editors to decide if the orthogonal representation of visual stimuli and any unknown (not necessarily cognitive) variable is still interesting, but their argument that the contextual signals represent "cognitive" variables should be substantially toned down and they should comment on "behavioral" variables as potential caveat.

We thank the reviewer for insisting on more precise characterization of behavioral strategies. We agree that low performance on incongruent no-go trials is a source of concern that requires adequate treatment, which we have addressed with quantitative models of behavior. These models show much performance advantage for context-aware models. Equally importantly, we have restructured the first part of the results to describe more intuitively the behavioral results.

Our responses to specifically address the two listed concerns of the reviewer are below.

In the first concern the reviewer suggests two points. First, the reviewer points out that the animal might have a tendency to lick even in cases when it shall withhold licking. This is indeed a potential strategy, corresponding to FOMO (fear of missing out). In the case of uncertainty about the right policy, the utility of the choice to lick might be more attractive because it only incurs a short timeout as opposed to not licking and missing out a potential reward (and also missing information about the availability of the reward, therefore missing an opportunity to learn about task rules). The critical question is whether the strategy is modulated by the task context or not. Based on the behavioral models, we argue that the behavioral strategy is modulated by context. The second point the reviewer makes is that the animal might still attend the auditory cue in the visual context. In this case we would like to point out that attending to the auditory cue would incur errors in the incongruent 'go' trials. Inspection of the example animal shows that out of 16 incongruent 'go' trials there is a single error, indicating that the animal was either

making the decision based on the visual modality or had a very high licking bias. In fact, out of the eight different trial types, the animal had the highest performance on the visual incongruent 'go' trials, only showing a single error. Therefore, making choices based on the opposite cue seems to be a less fitting explanation of the strategy characterizing the visual context. We have included the 'opposite' strategy in the lineup of the alternative behavioral models in Fig. 11.

The second concern of the reviewer is that increases in incongruent 'no-go' coincides with an increase in misses in congruent trials. Both in the visual and audio contexts there are two miss trials. This is actually similar to four other trial types where one or two errors are recorded. The timing of the two error trials do not seem to coincide with the start of a consistent (formerly high-performance) block, despite that the dynamics of running averages do seem to suggest such coincidences. Please note, however, that these drops in the average actually correspond to misses occurring a dozen trials later.

Similar observations can be made in all mice for both points.

To ensure that the methodology we used to identify consistent trials actually rules out a context-unaware licking strategy we performed validation by generating synthetic data. Our findings are shown on Fig. R1. Briefly, the analysis demonstrated that the criterion we used to select consistent trials does not accidentally classify an animal that displays a simple lick bias in incongruent trials as part of the analyzed cohort ($p < 0.005$).

We addressed movement-related confounds in a reply to point 2. Here we would like to emphasize the result in Fig. 3I,J, where we demonstrated an increased proportion of correctly predicted context based on neuronal responses in the consistent block. Certainly, this increase in context decoding accuracy can still arise from multiple causes, but if there is no evidence that behavioral difference is significant, the majority of difference should come from the internal state of the animal represented in neural firing patterns. In particular, based on the data we presented on the relationship between behavior and contextual modulation of neuron population data we do not believe that unaccounted factors that the reviewer proposed (stress or frustration) provide a parsimonious explanation for contextual modulation of population responses. Instead, we found that behavior that is more consistent with the rules of the task occurs when the contextual modulation is also enhanced (Fig. 3I,J). We highlighted the point on the increase in task execution and contextual modulation in the text (section Orthogonal representation for sensory stimuli and task context).

In summary, we agree with the reviewer that the *average* performance of the animals on incongruent 'no-go' trials is a factor that requires thorough analysis and the consideration of alternative strategies. The set of behavioral models we set up were consistent in that a contextual modulation of choices is present. We also agree with the reviewer that factors not direct consequences of the task structure (such as frustration, posture differences) can also contribute to the measured context signal. However, in the light of the above arguments and

analyses, we believe that the two concerns listed by the reviewer do not cast doubt on our point that the animals are performing the context shifting task although shifts in behavioral strategies occur during a session (causing occasional deterioration from the optimal strategy). In the manuscript we analyze the behavioral models on consistent trials. Such an analysis ensures that even though strategies might shift, periods when the animal is performing the right strategy are identified.

We do acknowledge that the way the behavioral part of the experiment was presented did not help the reader to navigate these concerns. Thus, besides adding new behavioral models, we have rewritten the first section of results completely. We briefly extended the discussion to include stress and frustration (section Cross-modal audiovisual task).

Also, the way how they calculated moving average of performance level in Figure 1F is misleading. For example, the red dotted line starts increasing even though the mouse is still making 100% false alarm decisions.

We use a 20-trial wide moving average, so when the edges of the averaging window encounter more successful trials as the window advances, the averaged curve will rise in the center (the point drawn for the central trial). We chose to use an 'acausal' version of moving average, which might lead to counterintuitive consequences such as increased average performance caused by a future decision. It is worth noting, that the same is true when the moving average trace starts to decline, it will appear to decline before the animal displays any errors. We believe that moving averages at this rate reflects the rate of behavioral changes, e.g. exploration cycles, which are consistently around 10-20 trials width. A more explicit statistical model for such shifts would rely on Hidden Markov Models to switch between strategies (e.g. Ashwood et al 2022, Nat Neurosci). Implementing this, however, proved to be unfruitful as the limited behavioral data collected in a session prevented us from directly using such a model in our analyses. Again, we believe that the observed time-dependent behavior is best described by a moving average. Note that the consistent blocks are further tested with context-aware and -unaware behavioral models, where each trial contributes to the likelihood individually (Fig 1I,J).

Finally, I don't get the authors' following statement in their response: "We divided the sessions into different types of epochs: 1. High performance epochs : Blocks of trials when animals performed well during incongruent trials during both visual and auditory attention sections. 2. Low performance epochs: Blocks of trials when the animal performed well during incongruent trials only during either visual or auditory attention trials, but not both." Given that visual and auditory context blocks are separated in time, how can such blocks of trials be defined? This is not the same as how high-performance trials were defined in the manuscript. Possibly I'm misunderstanding something...

We are sorry for the confusion a previous response to reviewers' text caused. 'Consistent' trials (formerly 'high performance' trials) are taken into consideration *together* in both contexts. If they

do not exist in one of the contexts, then we do not regard the other as 'consistent', either, even though >0.5 criteria is met in that single context.

9. The authors' response does not directly address the reviewer's point. At least to control for behavioral confound related to licking, the authors should redo the analysis by using only correct nogo trials and test if they still see the abrupt change.

We thank the reviewer for suggesting a control for the licking confound. We do not expect different across-time decoding patterns for two reasons. First, mean licking time is slightly after the 2 sec water availability, and mice begin licking in the first second of the stimulus presentation extremely rarely, let alone before stimulus starts. Licking may affect the change at stimulus offset, but should not affect the change at stimulus onset. Second, Fig 4I-N shows that context representation dynamics is more nuanced than always having an abrupt change at stimulus on and off. Furthermore we conjectured that resistance to perturbation depends on specific neurons, not necessarily within the sampled (measured) units of any given animal (note that Fig 4I-J displays all neurons from all animals). Confirming our expectations, by repeating the analysis Fig 4F,K,N on correct 'no-go' trials shows similar characteristics demonstrated on all trials (Fig. R3). Note, that the number of trials for decoding are now reduced by a factor of correct rate divided by two, cca. 30-40% of previously available trials, typically to 10-20 trials depending on mice. In conclusion, as we expected, lick controlled trials yield very similar results to using all trials, and that holds true in almost all mice individually.

Figure R3 Across time context accuracy decoding on correct 'no-go' trials.

REVIEWERS' COMMENTS

Reviewer #2 (Remarks to the Author):

All the concerns raised in the previous review have been addressed. There are a couple of minor suggestions:

1) The methods for the context-aware and context-unaware models are hard to follow. Both the description of these models and the fitting procedure to the data can be significantly improved both for understanding and the ability to replicate.

2) It would be useful to include Figure R1 as a supplementary figure in the manuscript.

Reviewer #4 (Remarks to the Author):

The authors have made substantial changes especially to control for the potential caveat that any behavioral differences could explain the contextual modulation of population activity, and the presentation of the findings shows significant improvement. Also, I feel that the orthogonal population coding of sensory and non-sensory information is interesting. However, my biggest concern on this study – mice have not learned the modality-switching task – still remains. In the revised manuscript, the authors compared behavioral models and concluded that the context-aware model better explained the mouse behavior during consistent blocks. However this modelling analyses are trivial since if trials are subselected from such time periods when task performance is relatively better and balanced across task types, the context-aware model must have a better explanation on the mouse behavior. I fear seriously that the authors might have (unintentionally) picked up time periods where mice performed the incongruent no-go task in the visual block “by chance” as consistent blocks from the very fluctuating performance levels as shown in Fig1F, especially given that Fig1G shows the duration of consistent blocks within a session are quite short in almost all animals assuming that all mice had the equal number of trials (the y axis should be shown as %). This cannot be ruled out by modeling behaviors during the consistent blocks. Therefore, the “context” that was decoded from V1 population activity is not necessarily related to cross-modal attentional switch but can just be easy vs difficult. This should be more clearly discussed as a caveat of this study.

Regarding to my concern about the anti-correlated performance of incongruent no-go and congruent go in the visual block, the example session in Fig1F is better to be replaced with that from other mice if other mice really don't show such a trend as the authors stated in their responses. Current figure will mislead the readers who are familiar with mouse behaviors. Likewise, although I understand the

rationale of using the running average, the current Fig1F clearly shows that the mouse just continue false-alarming during the first consistent block. This will cast doubt on the validity of the analysis, so better to be replaced if they can.

Response to reviewers

Reviewer 2

All the concerns raised in the previous review have been addressed.

Thank you for the feedback.

There are a couple of minor suggestions:

1) The methods for the context-aware and context-unaware models are hard to follow. Both the description of these models and the fitting procedure to the data can be significantly improved both for understanding and the ability to replicate.

Thank you for highlighting this. The text has been updated.

2) It would be useful to include Figure R1 as a supplementary figure in the manuscript.

Thank you for coining this. We have included an extended version of this figure as a new panel in Figure 1f.

Reviewer 4

The authors have made substantial changes especially to control for the potential caveat that any behavioral differences could explain the contextual modulation of population activity, and the presentation of the findings shows significant improvement. Also, I feel that the orthogonal population coding of sensory and non-sensory information is interesting.

Thank you for the reassuring words.

However, my biggest concern on this study – mice have not learned the modality-switching task – still remains. In the revised manuscript, the authors compared behavioral models and concluded that the context-aware model better explained the mouse behavior during consistent blocks. However this modelling analyses are trivial since if trials are subselected from such time periods when task performance is relatively better and balanced across task types, the context-aware model must have a better explanation on the mouse behavior. I fear seriously that the authors might have (unintentionally) picked up time periods where mice performed the incongruent no-go task in the visual block “by chance” as consistent blocks from the very fluctuating performance levels as shown in Fig1F, especially given that Fig1G shows the duration of consistent blocks within a session are quite short in almost all animals assuming that all mice had the equal number of trials (the y axis should be shown as %). This cannot be ruled out by modeling behaviors during the consistent blocks.

Thank you for consistently requiring assurances about the behavioral performance of animals. Assessment of the complete blocks of attend-audio and attend-visual trials instead of the selected trials is indeed important for showing that observed behavioral patterns are not produced by chance. We had performed an analysis addressing this issue that we had not included in the main text of the paper and had only presented in the previous round of response to reviewers in the part addressing Reviewer 2's concerns. We have now added this analysis to the main text as a new panel for Fig. 1. The analysis compares a baseline behavioral model to the behavior observed in our recorded animal. We investigated the probability with which consistent trials can occur under the baseline model. We calculated two separate probabilities: 1, irrespective of the size of the continuous block of consistent trials we assessed the probability of the length of the consistent block; 2, we calculated the probability of the continuous consistent block we observed. Both of the more relaxed (former) and the stricter (latter) measures indicated that the behavioral pattern we observed was unlikely under a model which produces consistent blocks by chance. We expanded on this in the main text of the manuscript.

Therefore, the “context” that was decoded from V1 population activity is not necessarily related to cross-modal attentional switch but can just be easy vs difficult. This should be more clearly discussed as a caveat of this study.

While our analysis provides evidence that consistent blocks are indeed corresponding to task-execution, which requires the representation of context, the direct correspondence between the context signal and task rules were not established in this study. As a consequence, the semantics of the signal identified with the context cannot be stated. We make this point clear in the discussion of the paper.

Regarding to my concern about the anti-correlated performance of incongruent no-go and congruent go in the visual block, the example session in Fig1F is better to be replaced with that from other mice if other mice really don't show such a trend as the authors stated in their responses. Current figure will mislead the readers who are familiar with mouse behaviors. Likewise, although I understand the rationale of using the running average, the current Fig1F clearly shows that the mouse just continue false-alarming during the first consistent block. This will cast doubt on the validity of the analysis, so better to be replaced if they can.

We strived to support claims about animal behavior with quantitative analysis. The trial-by-trial inspection of an animal certainly leaves space for alternative interpretations as the finite number of trials in a block can be easily overfitted. We believe that the running average is a concept that readers will be familiar with and therefore its properties will not be unknown to the readers. In summary, we opted to keep the example animal.